# Generalizable Opponent Exploitation in LLM Agents via Mixed Best-Responses Training

## Abstract

Opponent exploitation is a crucial capability for agents in competitive scenarios, allowing them to exploit weaknesses in opponent strategies. Large Language Model (LLM) based agents have demonstrated remarkable capabilities in strategic reasoning and adversarial decision-making. However, their ability to exploit diverse opponents, including those following suboptimal strategies, remains underexplored. In this work, we introduce **GOE-LLM** (Generalizable Opponent Exploitation with LLMs), a novel framework that leverages LLMs to learn opponent exploitation strategies through mixed best-response training in two-player zero-sum games. A Multi-Layer Perceptron (MLP) Profiler is pre-trained independently to analyze opponent behaviors and identify their strategic patterns. This profiling information is then utilized by a fine-tuned LLM Exploiter, trained with group relative policy optimization on a curated set of best-response strategies against heterogeneous opponents. To ensure stable training while enabling the resulting agent to generalize across a broad spectrum of opponents, we propose a Mixture-Best-Responses principle to guide the construction of training data. We evaluate GOE-LLM using various LLM sizes in Kuhn Poker, where it demonstrates strong exploitation against out-of-distribution opponents. Additionally, our method shows consistent performance and generalization trends in Leduc Hold'em Poker. We construct and compare different mixtures of training data to validate the effectiveness of the Mixture-Best-Responses principle, confirming its role in ensuring both stability and generalization. Extensive ablation studies further validate the contributions of each component to the overall performance. Our results highlight the potential of GOE-LLM for generalizable opponent exploitation and demonstrate the effectiveness of mixed best-response training in enhancing the adaptability of LLM agents.

## 1 Introduction

Opponent exploitation is a critical skill for agents in competitive scenarios, enabling them to adapt to specific opponents and exploit their weaknesses (Hoehn et al., 2005; Ganzfried & Sandholm, 2015; Liu et al., 2022). This skill is essential not only for achieving competitive performance but also for demonstrating advanced strategic reasoning.

Meanwhile, Large Language Models (LLMs) have recently demonstrated impressive capabilities as autonomous agents (Wang et al., 2024), excelling in strategic reasoning and decision-making within interactive domains such as multi-agent games (Zhang et al., 2024b; Duan et al., 2024; Huang et al., 2025; Light et al., 2023; 2025; Guo et al., 2024; Hu et al., 2024). Despite such rapid progress, however, the potential of LLMs for *opponent exploitation* has not yet been thoroughly explored. In fact, existing research on LLM-based game agents has primarily focused on two approaches: (1) in-context learning with carefully engineered prompts (Brown et al., 2020; Zhang et al., 2024a; Guan et al., 2024; Karten et al., 2025; Cui et al., 2025b; Xu et al., 2025b), and (2) direct policy learning methods using data generated from equilibrium or optimal strategies (Huang et al., 2024; Zhuang et al., 2025; Feng et al., 2023; Zhang et al., 2025; Wang et al., 2025; Xu et al., 2025a). While these methods have advanced the field, they tend to prioritize general strategic competence or equilibrium play (Zhuang et al., 2025). Consequently, there remains a notable gap between current approaches and the goal of developing LLM-based agents that can explicitly adapt to—and effectively exploit—the weaknesses of diverse opponents. This observation raises a central question:

*what would it take for LLM-based agents to move beyond equilibrium play and achieve effective and generalizable opponent exploitation?*

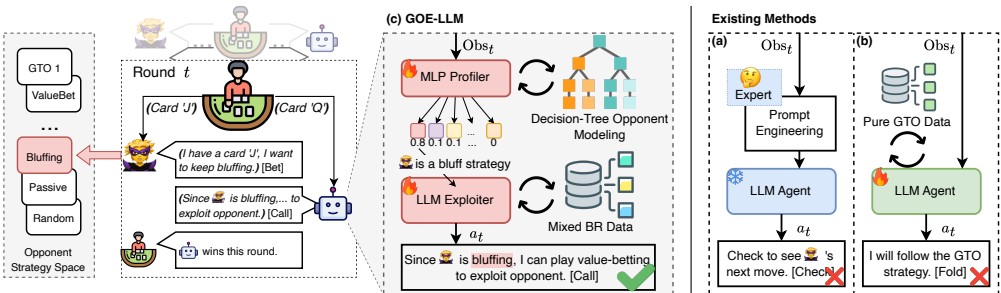

Figure 1: GOE-LLM Overview. In two-player zero-sum poker games, players take turns acting, and can exploit opponents deviating from optimal strategies to gain more chips. Existing research mainly focuses on (a) in-context learning with expert knowledge, which suffers from hallucinations, and (b) training on single optimal strategy data, which fails to exploit opponent weaknesses. We propose (c) GOE-LLM, where a top-level profiler classifies opponent strategies, and a bottom-level exploiter is trained on a mixture of best-response data to enhance adaptability to diverse opponents.

Motivated by this challenge, we propose **GOE-LLM** (Generalizable Opponent Exploitation with LLMs), which introduces a LLM-centered framework designed to enhance opponent exploitation capabilities, as illustrated in Figure 1. GOE-LLM integrates (i) a lightweight profiler that identifies strategic tendencies of opponents, and (ii) an LLM-based exploiter fine-tuned to generate best-responses against diverse opponents. Inspired by the mixture of opponents learning (Smith et al., 2023), we employ a mixed best-response training paradigm. First, we mix multiple best-response strategies in a balanced way, ensuring that the exploiter learns not only equilibrium strategies but also diverse best-response strategies. In particular, we carefully construct the mixture to prevent non-transitive dominance cycles (Tenney & Foster, 1976) between best-response strategies, thereby preserving training stability and enhancing generalization. Building on the mixed best-response data, we further adapt the Group Relative Policy Optimization(GRPO) algorithm (Shao et al., 2024) to fine-tune the LLM exploiter. By designing a fine-grained reward signal that incorporates opponent-aware reasoning, we encourage the model to explicitly consider opponent behavior in its decision process. This design also establishes a natural connection to the upper-level profiler, ultimately enhancing the exploiter to generalize and effectively exploit weaknesses across both familiar and previously unseen opponents. We conduct experiments in Kuhn Poker and Leduc Hold'em poker, showing that GOE-LLM not only achieves strong generalization to out-of-distribution opponents but also maintains stable performance across varying LLM sizes. Through comprehensive ablation studies, we further disentangle the contributions of each component and analyze the impact of different mixture training strategies on performance. Overall, our contributions are threefold: (1) we propose the LLM-based framework explicitly targeting opponent exploitation; (2) we develop a mixed best-response training principle that balances robustness and diversity for generalizable opponent exploitation, along with a corresponding opponent-aware policy optimization; and (3) we provide extensive empirical validation, highlighting the effectiveness of GOE-LLM in Kuhn poker and Leduc Hold'em poker, along with insights into the roles of various components.

## 2 RELATED WORK

Research on opponent exploitation has traditionally combined game theory and reinforcement learning. Classical approaches such as Independent Reinforcement Learning (InRL), Iterated Best Response (IBR), Double Oracle (DO), and Fictitious Play (FP) aim to compute optimal responses to opponents' strategies (Lanctot et al., 2017), but they often overfit to specific equilibria and generalize poorly to unseen adversaries. To handle non-transitive strategy cycles, Rectified Nash Response (PSROrN) was introduced to maintain diversity through ecological niches (Balduzzi et al., 2019). The Policy Space Response Oracles (PSRO) framework then unified DO and FP by iteratively expanding restricted games with empirical game-theoretic analysis (Bighashdel et al., 2024). Extensions include PSD-PSRO, which regularizes for population robustness (Yao et al., 2023a),

and SPSRO, which adapts meta-solvers and response learning dynamically (Li et al., 2024a). In large-scale domains like StarCraft II, league training frameworks integrate opponent modeling and goal-conditioned exploiters, boosting exploitation and adaptation in real time (Huang et al., 2023). Despite that, insufficient strategy diversity and overfitting remain central challenges.

More recently, large language models (LLMs) have been introduced into game environments as central components for opponent modeling and analysis, enabling more flexible and semantically grounded exploitation strategies (Hu et al., 2024). Current LLM-based methods primarily rely on prompting or lightweight fine-tuning, but they have yet to systematically address opponent exploitation. Prior work shows that LLM-based agents can adapt to game environments by using carefully designed prompting strategies to follow rules and generate coherent action sequences (Zhang et al., 2024a; Karten et al., 2025). Likewise, fine-tuning on curated datasets or synthesized decision-making trajectories can improve domain-specific performance. For example, the POKERBENCH benchmark demonstrates that although frontier models like GPT-4 perform poorly in poker, fine-tuning yields modest gains but still falls far short of human expert play (Zhuang et al., 2025). Similarly, Mastermind-Dou/Go fine-tune models on structured game data (Wang et al., 2025), while PokerGPT (Huang et al., 2024) uses RLHF and prompt-engineered trajectories to adapt lightweight LLMs to Texas Hold'em. The ICE framework attempts in-context opponent exploitation through trajectory-based RL, but it struggles with dynamic opponents and requires processing full historical trajectories (Li et al., 2024b; Shi et al., 2024). Existing methods, largely based on trajectory imitation or reward optimization, enhance consistency but fail to capture opponent-specific weaknesses. This limitation is most acute in imperfect-information games, where adaptive exploitation is essential.

Inspired by the ToMAP framework (Han et al., 2025), which strengthens persuasion agents with counterclaim prediction and MLP-based opponent modeling, we extend it to adversarial game settings. We propose a mixture-of-best-response principle to stabilize LLM mixture policy training and integrate an MLP profiler to model unseen opponents' behavioral tendencies, enabling adaptive strategy adjustment. This design combines ToMAP's opponent-awareness with game-theoretic exploitation, addressing instability and generalization limits in current LLM-based agents.

## 3 METHODOLOGY

In this section, we introduce GOE-LLM, our proposed generalizable opponent exploitation framework for LLM agents. Our framework is organized into two hierarchical layers: opponent modeling followed by opponent exploitation. Specifically, we employ a multilayer perceptron (MLP) as the opponent profiler, which dynamically classifies the opponent based on their recent behavior tree. The classification is translated into a language-based description, which is then passed to the LLM Exploiter, aimed at exploiting the identified opponent. Before detailing these two components, we first present the formal definition of the game setting for LLM-based agents.

### 3.1 PRELIMINARIES

**Two-player zero-sum imperfect-information extensive-form game.** An imperfect-information extensive-form game(IIEFG) $G = (N, A, H, Z, \chi, \rho, \sigma, u, \mathcal{I})$ describes a sequential interaction among $n$ players (Liu et al., 2022). $N = \{1, \ldots, n\}$ is a finite set of players, and $c$ denotes the chance player modeling exogenous randomness. $H$ is the set of non-terminal decision nodes, $Z$ is the set of terminal nodes (leaves). The set of all possible actions is $A$, and $\chi : H \to 2^A$ assigns to each decision node $h \in H$ the set of legal actions $\chi(h)$. A player function $\rho : H \to N \cup \{c\}$ assigns to each decision node $h$ the player (or chance) who acts at that node. $\sigma_c$ is the fixed, commonly known stochastic policy of the chance player and $u = (u_1, \ldots, u_n)$ is the utility function, where $u_i : Z \to \mathbb{R}$ specifies the payoff of player $i$ at each terminal node. $I = (I_1, \ldots, I_n)$ is the collection of information sets, where each $I_i = \{I_{i,1}, \ldots, I_{i,k_i}\}$ is a partition of the decision nodes of player $i$. If two nodes $h, h'$ belong to the same information set $I_{i,j}$, then $\rho(h) = \rho(h') = i$ and $\chi(h) = \chi(h')$. We use $I(h)$ to denote the infoset containing node $h$. The strategy of player $i$ is $\sigma_i : I_i \to \Delta(A)$, where $\Delta(A)$ is the set of probability distributions over $A$. A strategy profile is $\sigma = (\sigma_1, \ldots, \sigma_n)$. And the expected utility of player $i$ under strategy profile $\sigma$ is denoted by $u_i(\sigma) = u_i(\sigma_i, \sigma_{-i})$, where $\sigma_{-i}$ is the strategy profile of all players except player $i$. For two-player zero-sum IIEFGs, we have $n = 2$ and $u_1 + u_2 = 0$. We denote the strategies of player 1 and player 2 are $\sigma_1$ and $\sigma_2$, respectively. The value of the game is defined as $v = \max_{\sigma_1} \min_{\sigma_2} u_1(\sigma_1, \sigma_2)$. A best response

strategy for player 1 against opponent strategy $\sigma_2$ is defined as $\text{BR}_1(\sigma_2) = \arg\max_{\sigma_1} u_1(\sigma_1, \sigma_2)$. A strategy profile $\sigma^* = (\sigma_1^*, \sigma_2^*)$ is a Nash equilibrium if $u_1(\sigma_1^*, \sigma_2^*) \geq u_1(\sigma_1, \sigma_2^*)$ for all $\sigma_1$ and $u_2(\sigma_1^*, \sigma_2^*) \geq u_2(\sigma_1^*, \sigma_2)$ for all $\sigma_2$. In two-player zero-sum games, a Nash equilibrium strategy is also a minimax strategy, i.e., $\sigma_1^* = \text{BR}_1(\sigma_2^*)$ and $\sigma_2^* = \text{BR}_2(\sigma_1^*)$.

**Game-playing with LLM agents.** Since LLMs are not explicitly designed to model game states and strategies, we leverage their strong instruction-following capabilities by integrating the game rules and infoset information into the prompt. This enables the LLM to understand the basic rules and current state of the game, allowing it to make valid decisions without explicitly modeling the game state and strategy. Formally, at each decision node $h$, the LLM agent receives $\text{prompt}_h$ that includes the game rules, the current infoset $I(h)$, and the history of actions taken so far. The LLM then generates an action $a \in \chi(I(h))$ based on this prompt: $a \sim \sigma_{\text{LLM}}(I(h)) = f(\pi_\theta(\text{prompt}_h))$, where $\pi_\theta$ is the LLM parameterized by $\theta$, and $f$ extracts the action from the LLM's output.

### 3.2 MLP Opponent Profiler

The MLP profiler is pre-trained to provide opponent information to the LLM exploiter. We collect opponent data from predefined opponent types to train an MLP classifier, which maps the opponent's behavior tree over the last $k$ games to a discrete opponent type space. The opponent type is then translated into a language-based description. It is important to note that the choice of $k$ varies across different environments and opponent definitions. A small $k$ may fail to capture the opponent's behavior patterns, while a large $k$ may overlook recent weaknesses. In Kuhn Poker and Leduc Hold'em poker, we set $k = 10$ and $k = 50$, respectively. Due to space constraints, we provide more training details and choice of $k$ in the Appendix A.4, along with visualization results.

### 3.3 LLM Opponent Exploiter

We first describe the training procedure of the LLM exploiter. Its objective is to learn a robust policy that adapts to different opponent types and approximates their best responses. To achieve this, we propose a Mixture-of-Best-Responses Principle that balances training stability and generalization. Based on this principle, we extend the GRPO algorithm (Shao et al., 2024) with a fine-grained opponent-aware reward design to optimize the LLM exploiter, shown in Figure 2.

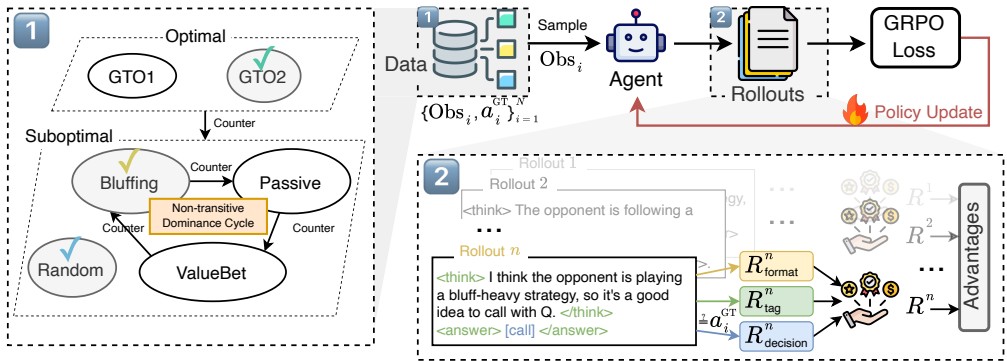

Figure 2: The training procedure of LLM Opponent Exploiter. There are two key components: (1) Mixture-of-Best-Responses Principle, which guides the selection of training data, ensuring a diverse yet non-cyclic counter relation among strategies; (2) Opponent-Aware Policy Optimization, which optimizes the LLM exploiter using GRPO with a fine-grained opponent-aware reward structure.

#### 3.3.1 Mixture-of-Best-Responses Principle

Training LLM game agents directly on equilibrium strategies often leads to overfitting (Lanctot et al., 2017), making it difficult to adapt to diverse opponents. Conversely, learning to counter all possible opponent strategies can result in unstable training.

Let the strategy space of two-player zero-sum IIEFG be $\Sigma$. For two strategies $\sigma, \sigma' \in \Sigma$, we say that $\sigma$ *counters* $\sigma'$, written as $\sigma \succ \sigma'$, if $\tilde{u}(\sigma, \sigma') > 0$, where $\tilde{u}(\sigma, \sigma')$ denotes the symmetrized payoff

of strategy $\sigma$ against $\sigma'$, obtained by averaging its expected payoff over both player positions in the zero-sum game. We Let $\Sigma_i, \Sigma_j \subseteq \Sigma$ denote two disjoint subsets (strategy clusters). We say that $\Sigma_i$ *counters* $\Sigma_j$, written as $\Sigma_i \succ \Sigma_j$, if

$$\forall \sigma \in \Sigma_i, \ \forall \sigma' \in \Sigma_j: \quad \sigma \succ \sigma' \Leftrightarrow \tilde{u}(\sigma, \sigma') > 0.$$

For any strategy $\sigma \in \Sigma$, by definition of best response in two-player zero-sum games, we have $\tilde{u}(\mathrm{BR}(\sigma), \sigma) > 0$, which implies that $\mathrm{BR}(\sigma)$ counters $\sigma$. We define its *counter set* as $\mathrm{Counter}(\sigma) := \{ \sigma' \in \Sigma \mid \sigma' \succ \sigma \}$, i.e., the set of all strategies that counter $\sigma$.

**Cyclic counter relation.** A strategy set $\Sigma' \subseteq \Sigma$ is said to exhibit a *cyclic counter relation* if there exists a subset $\{\sigma^1, \sigma^2, \ldots, \sigma^k\} \subseteq \Sigma'$ that forms a *non-transitive dominance cycle*, i.e.,

$$\sigma^1 \succ \sigma^2, \quad \sigma^2 \succ \sigma^3, \quad \ldots, \quad \sigma^{k-1} \succ \sigma^k, \quad \sigma^k \succ \sigma^1.$$

Training dataset $\mathcal{D}$ is generated from interactions with a collection of profiles $\{(\sigma^i_{\mathrm{oppo}}, \sigma^i_{\mathrm{agent}})\}_{i=1}^m$, where $\sigma^i_{\mathrm{oppo}} \in \Sigma$ denotes the opponent's strategy and $\sigma^i_{\mathrm{agent}} \in \Sigma$ is the agent's, chosen such that

$$\sigma^i_{\mathrm{agent}} \in \mathrm{Counter}(\sigma^i_{\mathrm{oppo}}) \cup \mathrm{BR}(\sigma^i_{\mathrm{oppo}}).$$

We say that $\mathcal{D}$ satisfies the *Mixture-of-Best-Responses Principle* if the set of agent strategies

$$\{\sigma^1_{\mathrm{agent}}, \sigma^2_{\mathrm{agent}}, \ldots, \sigma^m_{\mathrm{agent}}\}$$

does not exhibit a cyclic counter relation. Intuitively, this Principle requires that the mixture of best-response strategies used for training is free from cyclic counter relation, thereby ensuring both training stability and the potential for generalization across diverse opponents. We show the ablation results of different mixture principles in Section 4.3.

### 3.3.2 OPPONENT-AWARE POLICY OPTIMIZATION

To optimize the LLM exploiter, we adopt GRPO, a reinforcement learning algorithm designed to eliminate the need for an explicit value function while stabilizing training in reasoning tasks. Unlike Proximal Policy Optimization (PPO), which relies on estimating token-level advantages using a learned critic, GRPO leverages a group-based normalization mechanism to estimate the relative quality of generated responses.

Formally, given a prompt $q$ (corresponding to an infoset in the game) and a group of $G$ sampled responses $\{o_i\}_{i=1}^G$ generated from the old policy $\pi_{\theta_{\mathrm{old}}}$, the reward of each response is computed as $R_i$. The relative advantage of response $i$ is estimated by normalizing rewards within the group:

$$\hat{A}_i = \frac{R_i - \mathrm{mean}(\{R_j\}_{j=1}^G)}{\mathrm{std}(\{R_j\}_{j=1}^G)}.$$

This formulation provides a variance-reduced estimate of relative quality, encouraging the policy to assign higher probability mass to actions leading to better-than-average outcomes while suppressing worse ones. The optimization objective of GRPO is given by:

$$J_{\mathrm{GRPO}}(\theta) = \mathbb{E}_{(q,a)\sim\mathcal{D}, \{o_i\}_{i=1}^G \sim \pi_{\theta_{\mathrm{old}}}(\cdot|q)} \left[ \frac{1}{G} \sum_{i=1}^G \frac{1}{|o_i|} \sum_{t=1}^{|o_i|} \left( \min(r_{i,t}(\theta)\hat{A}_i, \mathrm{clip}\hat{A}_i) - \beta D_{\mathrm{KL}} \right) \right]. \quad (1)$$

$$\mathrm{clip} = \mathrm{clip}(r_{i,t}(\theta), 1 - \epsilon, 1 + \epsilon), \quad D_{\mathrm{KL}} = D_{\mathrm{KL}}(\pi_\theta \| \pi_{\mathrm{ref}}).$$

where the importance ratio is defined as

$$r_{i,t}(\theta) = \frac{\pi_\theta(o_{i,t} \mid q, o_{i,<t})}{\pi_{\theta_{\mathrm{old}}}(o_{i,t} \mid q, o_{i,<t})}.$$

Here, $\epsilon$ is the clipping parameter that controls the trust region of policy updates, and $\beta$ regulates the KL divergence between the updated policy $\pi_\theta$ and a frozen reference policy $\pi_{\mathrm{ref}}$.

**Opponent-aware reasoning reward.** Rewards $R_i$ consist of three components:

- Format reward $R_{\text{format}} \in [0, 1]$: Rewards the agent for adhering to a structured output format, `<think>...</think><answer> [ACTION] </answer>`. The `<think>` tag encapsulates the reasoning process and the `<answer>` tag indicates the final action decision. Only outputs that strictly follow this format will receive a reward of $R_{\text{format}} = 1$, otherwise $R_{\text{format}} = 0$.

- Tag reward $R_{\text{tag}} \in [0, 1]$: Based on the specified format, the reply should include `<think>`, `</think>`, `<answer>`, and `</answer>`. Each tag must appear exactly once to earn a reward of 0.25 added to $R_{\text{tag}}$. Missing or repeated tags receive no reward for that tag (Han et al., 2025).

- Decision reward $R_{\text{decision}} \in [0, 1]$: Correction reward based on the accuracy of the final action decision. If the `[ACTION]` matches the ground truth, $R_{\text{decision}} = 1$; otherwise, $R_{\text{decision}} = 0$.

This composite reward structure incentivizes the agent to not only make correct decisions but also to articulate its reasoning process clearly, enhancing interpretability and consistency. Formally,

$$R_i = R_{\text{decision}} + \alpha_{\text{tag}} R_{\text{tag}} + \alpha_{\text{format}} R_{\text{format}},$$

where $\alpha_{\text{tag}}, \alpha_{\text{format}}$ are hyperparameters that balance the contributions of each component.

# 4 EXPERIMENTS

In this section, we evaluate the effectiveness of GOE-LLM, our proposed LLM-based Generalizable Opponent Exploitation framework, in enhancing the opponent exploitation performance in two-player zero-sum games. We conduct experiments on Kuhn poker and Leduc Hold'em, comparing GOE-LLM with several baseline methods. We also perform ablation studies to assess the contributions of different components of GOE-LLM. Before presenting the results, we first describe the experimental setup, including the environments, baselines, and evaluation metrics.

## 4.1 EXPERIMENTAL SETUP

**Environments.** We implement GOE-LLM on two classic two-player zero-sum poker games: Kuhn poker and Leduc Hold'em (Southey et al., 2005). Both games have an action space consisting of check, bet, call, raise, and fold, and involve multiple rounds. We use the environment implementation from the textarena (Guertler et al., 2025).

To design diverse opponent strategies, we predefine six types of opponent strategies: GTO, Random, Bluff, ValueBet, Passive, and Aggressive. The GTO strategy for Kuhn poker can be found in Kuhn (2016), while for Leduc Hold'em, we compute the GTO strategy using the CFR algorithm (Zinkevich et al., 2007). We provide detailed descriptions of opponent strategies in Appendix A.1, A.2.

**Baselines.** We compare GOE-LLM with several baseline methods, including rule-based agents, LLM-based agents without policy optimization, and fine-tuned LLM agents. Unless otherwise specified, LLM-based methods use Qwen2.5-3B as the base model. All LLM-based agents use the same prompt described in Appendix A.3. The training hyperparameters are summarized in Appendix A.5. In the ablation studies, we compare different sizes of LLMs, including Qwen2.5-1.5B, Qwen2.5-7B. We set $\alpha_{\text{tag}}$ and $\alpha_{\text{format}}$ to 0.1, following the setting in Han et al. (2025). The baselines are as follows:

- **Rule-based Agents**: We implement several rule-based agents with predefined strategies:
  - **Random Agent**: An agent that selects available actions uniformly at random.
  - **GTO Agent** (Kuhn, 2016; Zinkevich et al., 2007): An agent that follows a precomputed optimal strategy for the game.
  - **Best-Response Agent**: An agent that selects the best response strategy against the hidden opponent, assuming full knowledge of the opponent. This serves as an upper bound.
- **LLM Agent** (Yao et al., 2023b): An agent that uses a large language model to make decisions based on the current observation without opponent modeling(OM) information from profiler.
- **LLM+OM Agent**: An agent that uses a large language model with extra OM information, but without parameter fine-tuning.

- **GTO-LLM Agent** (Zhuang et al., 2025): An agent that is fine-tuned on pure GTO strategy data and makes decisions without OM information during inference.

**Evaluation Metrics.** We construct a total of 24 different opponent strategies from the six predefined types, including 21 exploitable strategies and 3 optimal (GTO) strategies. In Leduc Hold'em, we construct 14 different opponent strategies from five predefined types. The details of the opponent strategies are summarized in Table 1. To evaluate the performance of each agent, we let each agent play 3000 hands against each opponent strategy in Kuhn poker and 600 hands in Leduc Hold'em. The evaluation metrics include the average win chips per hand and the win rate (Southey et al., 2005). Implementation details of the predefined strategies can be found in Appendix A.1.

Table 1: Opponent strategy categories, counts, exploitability, and whether seen during training in Kuhn poker and Leduc Hold'em.

|  (a) Kuhn poker. | | | | | (b) Leduc Hold'em. | | | |
| --- | --- | --- | --- | --- | --- | --- | --- | --- |
| **Type** | **#Num** | **Exploitable** | **In Training** | | **Type** | **#Num** | **Exploitable** | **In Training** |
| Random | 1 | Yes | Yes | | Random | 1 | Yes | Yes |
| GTO | 3 | No | Yes | | GTO | 1 | No | Yes |
| Bluff | 8 | Yes | Yes | | Tight | 3 | Yes | Yes |
| Value | 4 | Yes | No | | Loose | 3 | Yes | No |
| Passive | 4 | Yes | No | | Passive | 3 | Yes | No |
| Aggressive | 4 | Yes | No | | Aggressive | 3 | Yes | No |
| **Total** | **24** | - | - | | **Total** | **14** | - | - |

Each agent plays 3000 games against each opponent strategy, and we report the win chips and the win rate as the evaluation metrics.

## 4.2 MAIN RESULTS

Table 2 and Table 3 summarize the performance of GOE-LLM in Kuhn poker and Leduc Hold'em, respectively. GOE-LLM shows strong opponent exploitation ability compared to all baseline methods. Facing both seen and unseen various opponents, GOE-LLM consistently achieves additional chips per hand compared to the expected value of a GTO strategy, demonstrating its effectiveness in exploiting suboptimal opponents. Evaluation results about win rate are provided in Appendix B.1.

**GOE-LLM is a strong opponent exploiter.** As shown in Table 2, GOE-LLM outperforms all baseline methods across all opponent strategies. GOE-LLM has a superior performance, achieving an average win of 0.021 chips per hand as P0 and 0.133 chips per hand as P1, which most closely approaches the idealized Best-Response agent's performance of 0.142 and 0.181. Similarly, in Leduc Hold'em, GOE-LLM also achieves the best performance, with an average win of 0.083 chips per hand as P0 and 0.034 chips per hand as P1. These indicate that GOE-LLM can effectively adapt its intrinsic strategy against different opponents to exploit suboptimal strategies.

**Training on mixed data promotes generalization.** Compared to training solely on pure GTO data, training on mixed data enables better generalization to out-of-distribution (OOD) opponents. As shown in Table 2, GTO-LLM plays a standard GTO strategy against the GTO opponent, achieving an average return of -0.055 chips per hand as P0 and 0.052 chips per hand as P1, which is close to the equilibrium value of Kuhn poker. However, its performance drops significantly against Random and Aggressive strategies, indicating that the LLM may have merely memorized the equilibrium strategy without truly learning it. GOE-LLM achieves returns above the equilibrium value against both seen and unseen opponents, demonstrating its strong generalization ability.

**The Vanilla LLM is a weak opponent exploiter.** The base LLM without fine-tuning exhibits limited opponent exploitation capabilities. For Kuhn poker agent, the LLM agent achieves an average return of -0.077, -0.073 chips per hand. By incorporating additional opponent modeling information, the LLM+OM agent achieves an average return of -0.100, -0.114 chips per hand, showing no improvement. This phenomenon is similar in the Leduc Hold'em, where LLM and LLM+OM agents achieve average returns of -0.577, -0.343 chips per hand and -0.790, -0.285 chips per hand, respectively. This indicates that the base LLM cannot effectively utilize opponent information for exploitation, highlighting the necessity of fine-tuning on relevant data.

Table 2: Performance comparison of GOE-LLM with baseline methods in Kuhn poker. Each row represents an agent, each column represents an opponent type, and each cell shows the average return per hand when the agent plays as P0 or P1. The best performance for P0 is highlighted in **bold**, while that for P1 is highlighted with a yellow background. BR* denotes an idealized best-response agent that selects the optimal counter strategy against each opponent.

| Agent | | Opponent (as $P_{opp}$) | | | | | | |
|---|---|---|---|---|---|---|---|---|
| Method | Role | Random | GTO(s) | Bluff(s) | Value(s) | Passive(s) | Agg.(s) | Average |
| Random | P0 | 0.111 | −0.171 | −0.221 | −0.303 | −0.146 | −0.136 | −0.145 |
| | P1 | −0.150 | −0.106 | −0.279 | −0.125 | −0.020 | −0.143 | −0.137 |
| GTO | P0 | 0.133 | **−0.051** | −0.064 | −0.052 | −0.065 | −0.050 | −0.025 |
| | P1 | 0.167 | 0.063 | 0.060 | 0.046 | 0.053 | 0.070 | 0.076 |
| LLM | P0 | **0.317** | −0.146 | −0.154 | −0.250 | −0.148 | −0.081 | −0.077 |
| | P1 | 0.054 | −0.109 | −0.166 | −0.211 | −0.063 | 0.060 | −0.073 |
| LLM+OM | P0 | 0.222 | −0.160 | −0.178 | −0.188 | −0.086 | −0.212 | −0.100 |
| | P1 | −0.085 | −0.143 | −0.219 | −0.091 | −0.035 | −0.112 | −0.114 |
| GTO-LLM | P0 | 0.010 | −0.055 | −0.035 | **0.145** | **0.043** | −0.117 | −0.001 |
| | P1 | 0.074 | 0.052 | 0.021 | 0.125 | 0.019 | −0.163 | 0.021 |
| GOE-LLM | P0 | 0.124 | −0.054 | **−0.028** | 0.073 | −0.015 | **0.024** | **0.021** |
| | P1 | 0.212 | 0.058 | 0.219 | 0.115 | 0.093 | 0.101 | 0.133 |
| BR* | P0 | 0.509 | −0.063 | 0.095 | 0.147 | 0.081 | 0.084 | 0.142 |
| | P1 | 0.415 | 0.052 | 0.243 | 0.126 | 0.125 | 0.127 | 0.181 |

Table 3: Performance comparison of GOE-LLM with baseline methods in Leduc Hold'em. Each row represents an agent, each column represents a specific opponent strategy.

| Agent | | Opponent (as $P_{opp}$) | | | | | | | | | | | | | |
|---|---|---|---|---|---|---|---|---|---|---|---|---|---|---|---|
| Method | Role | Random | GTO | Tight1 | Tight2 | Tight3 | Loose1 | Loose2 | Loose3 | Passive1 | Passive2 | Passive3 | Agg.1 | Agg.2 | Agg.3 | Average |
| LLM | P0 | 0.263 | −0.063 | 0.092 | −0.054 | 0.529 | −0.975 | −1.208 | −1.850 | **0.433** | −0.375 | −0.408 | −1.113 | −1.429 | −1.917 | −0.577 |
| | P1 | −0.463 | 0.121 | 0.379 | −0.063 | 0.242 | −0.425 | −0.292 | −1.083 | −0.529 | −0.242 | −0.525 | −0.683 | −0.754 | −0.488 | −0.343 |
| LLM+OM | P0 | 0.007 | −0.618 | **0.727** | **0.815** | 0.792 | −1.760 | −2.108 | −2.438 | 0.362 | **0.108** | **0.337** | −2.423 | −2.413 | −2.440 | −0.790 |
| | P1 | −0.285 | 0.115 | 0.577 | 0.917 | 0.767 | −1.358 | −1.012 | −1.083 | 0.533 | 0.582 | 0.248 | −1.362 | −1.495 | −1.197 | −0.285 |
| GOE-LLM | P0 | **0.268** | **0.042** | −0.098 | −0.105 | −0.120 | **0.118** | **0.277** | **0.082** | 0.088 | −0.167 | 0.062 | **0.185** | **0.190** | **0.347** | **0.083** |
| | P1 | 0.203 | 0.082 | −0.063 | −0.093 | 0.175 | 0.143 | −0.035 | 0.287 | 0.043 | 0.093 | 0.060 | −0.128 | −0.265 | −0.027 | 0.034 |

## 4.3 ABLATION STUDIES

**Data Mixture Strategies for LLM Training.** To evaluate the impact of different data mixture strategies on the training process and final performance, we design three data mixture strategies for comparison experiments: 1) GOE(only GTO), only using GTO data for training; 2) GOE, training data includes multiple strategies while satisfying the mixture principle; 3) GOE(+NTDC), adding opponent strategy data with Non-Transitive Dominance Cycle on the basis of GOE. Figure 3 shows the main training results. More detailed training curves are presented in Appendix C.2.

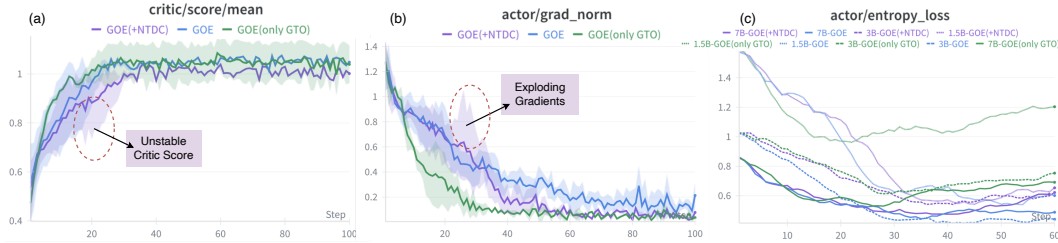

Figure 3: Ablation study on different data mixture strategies based on various LLM sizes.

From Figure 3(a), we observe that GOE(+NTDC) exhibits instability in the early stages of training due to the presence of Non-Transitive Dominance Cycles, resulting in significant fluctuations, which gradually converges after 40 training steps. Our GOE method, by avoiding the existence of cyclic counter relations, enables the model to be more stable during training, converging after about 25 steps. GOE(only GTO) converges the fastest, as it only needs to learn one best response strategy.

A similar trend can be observed in the gradient norm in Figure 3(b), where GOE(+NTDC) has exploding gradients in the early stages of training, further validating the positive effect of the mixture principle on training stability. Figure 3(c) shows that models of different sizes based on the GOE method can steadily reduce the entropy loss, indicating that the model gradually learns opponent exploitation strategies. After convergence, the models maintain an entropy above 0.4, indicating that the models retain a certain level of strategy diversity (Cui et al., 2025a).

**Contribution of MLP Profiler.** To assess the contribution of the MLP Profiler in GOE-LLM, we conduct an ablation study by removing the profiler, as GOE-LLM(w/o Profiler). As shown in Figure 4, GOE-LLM (w/o profiler) performs well against seen opponents. With opponent information from the MLP Profiler, GOE-LLM achieves substantial improvements against unseen opponents, while preserving its performance on seen ones. This improvement is mainly attributed to the high opponent-type identification accuracy, which exceeds 96% across all types and enables the exploiter to adjust its strategy more efficiently. Table 5 in Appendix A.4 reports the accuracy across opponent types. We also test the profiler under different player positions; details are in Appendix C.1.

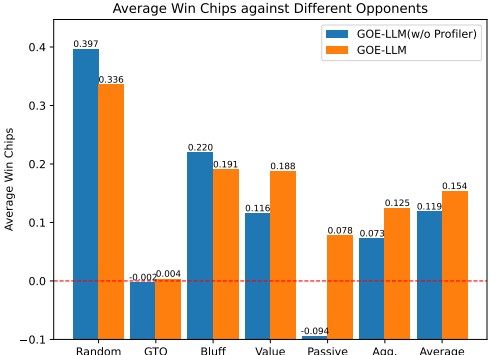
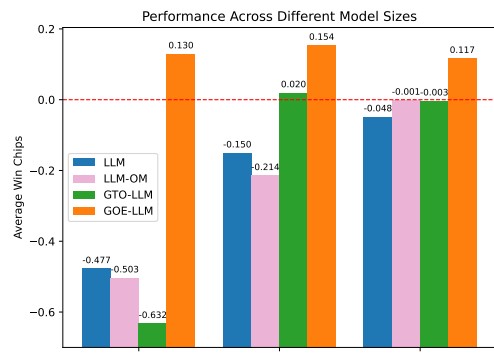

Figure 4: Average win chips per hand of GOE-LLM with and without the MLP Profiler.

Figure 5: Average win chips per hand with different LLM sizes, compared to baselines.

**Various LLM Sizes.** Figure 5 presents the performance of GOE-LLM with different LLM sizes, including Qwen2.5-1.5B, Qwen2.5-3B, and Qwen2.5-7B. Observing the results of LLM-based methods, the overall trend is that as the model size increases, the performance gradually improves. For the LLM+OM method, adding extra opponent information for smaller models not only fails to enhance strategy capability but also degrades performance; however, it does improve for the 7B model. This may arise from smaller models' limited ability to understand the environment and follow instructions, making the extra information more likely to hinder rather than help their decision-making. The GTO-LLM method enables the model to learn a good equilibrium strategy, stabilizing around 0. Our method, GOE-LLM, achieves good performance across different model sizes, reaching a peak performance of 0.154 chips per hand. The results demonstrate that our method can effectively enhance opponent exploitation capabilities across different model sizes.

## 5 CONCLUSION

GOE-LLM, a novel framework, enables LLM agents to perform opponent exploitation in two-player zero-sum imperfect-information games. By introducing a mixture of best-response data, we not only stabilize the learning of mixed strategies for the LLM exploiter but also avoid collapsing to a single equilibrium strategy, thereby generating diverse best-response strategies. The lightweight MLP profiler is trained to identify opponent types from recent behaviors of opponent and translate them into language-based descriptions for the LLM exploiter. This enables the LLM exploiter to adapt to unseen or out-of-distribution opponent types. We validate the effectiveness of the GOE-LLM framework in Kuhn poker and Leduc Hold'em, highlighting the importance of the mixture-best-responses principle in maintaining stable training and diverse strategies for the LLM exploiter. A current limitation is that GOE-LLM has not yet been assessed in more realistic general-sum or mixed cooperative–competitive domains, and exploring such settings represents an important direction for future work. Overall, our findings demonstrate the potential of leveraging LLM-based agents for generalizable opponent exploitation in more complex imperfect-information games.

ETHICS STATEMENT

This work does not involve human subjects, sensitive personal data, or proprietary information. The poker environments used (Kuhn poker and Leduc Hold'em poker) are standard benchmarks and do not raise ethical concerns. All datasets are synthetically generated and publicly shareable. We are not aware of any direct societal risks or negative downstream applications of our study. We have adhered to the Ethics throughout the preparation and submission of this work.

REPRODUCIBILITY STATEMENT

In the Supplementary Material, we provide the code for data generation, datasets, training and evaluation scripts, as well as detailed experimental results to facilitate verification and replication of our findings. The experimental environments for Kuhn poker and Leduc Hold'em poker are adapted from `textarena`: `https://github.com/LeonGuertler/TextArena`. Our implementation of the CFR algorithm follows an open-source reference: `https://github.com/tansey/pycfr`. The LLMs used in this work are downloaded from HuggingFace, fine-tuned with the `verl` library(`https://github.com/volcengine/verl`), and evaluated using `vllm`(`https://github.com/vllm-project/vllm`).

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

CONTENTS OF APPENDIX

APPENDIX

# A  IMPLEMENTATION DETAILS

## A.1  STRATEGIES

**Kuhn Poker Agents.**

In our implementation, we build a series of rule-based strategy agents based on the game structure of Kuhn Poker. The core idea is to use predefined probability tables conditioned on card strength, player position, and action history to decide whether to bet, call, fold, or check, thereby simulating players with different styles. Each type of agent is controlled by a small set of parameters: GtoAgent uses a parameter $\alpha$ to adjust the bluffing frequency with weak cards, where $\alpha = 1/3$ approximates the equilibrium strategy, smaller values represent more conservative play, and larger values indicate more aggressive play; BluffAgent extends $\alpha$ to higher ranges and allows an additional parameter to control bluffing tendencies after the opponent checks, thereby modeling over-bluffing strategies; ValueAgent and PassiveAgent parameterize the degree of value betting and passive folding, respectively, to represent solid or conservative styles; while AggressiveAgent uses its parameter to determine bluffing frequency with weak cards, modeling highly aggressive play. We also provide natural language opponent descriptions for prompt conditioning. Overall, the strategies exhibit a typical non-transitive countering relationship: Aggressive (bluff-heavy) strategies can exploit Passive strategies, Passive strategies can counter Value strategies, and Value strategies can effectively counter Aggressive strategies.

**Leduc Hold'em Agents.**

In the case of Leduc Hold'em, we similarly design a set of rule-based opponent strategies to capture diverse and contrasting playing styles. The baseline LeducGTOAgent follows approximate equilibrium strategies derived from PyCFR, balancing value betting and bluffing across different rounds. On top of this baseline, we introduce parameterized variants: LeducTightAgent emphasizes conservative play by drastically reducing betting or bluffing frequencies with weaker hands, focusing only on strong holdings; LeducLooseAgent represents the opposite style, frequently entering pots and bluffing with weak hands, thereby creating exploitable over-aggression; LeducAggressiveAgent systematically increases betting and raising frequencies regardless of card strength, aiming to apply constant pressure; while LeducPassiveAgent reduces proactive betting and relies mainly on checking and calling, reflecting a cautious and defensive style. We also include a LeducRandomAgent as a baseline without strategic structure. Each agent is controlled by simple parameters (e.g., tightness, looseness, aggression, passiveness) that adjust the relative weights of betting, calling, folding, and checking. For clarity in evaluation, we additionally provide natural language descriptions of these strategies to guide prompt conditioning. Overall, these strategies represent interpretable stylized opponents in Leduc Hold'em, where tight strategies can be exploited by bluffing, loose strategies can be punished by calling more often, aggressive styles are countered by defensive value play, and passive styles are vulnerable to pressure and frequent betting.

## A.2  DATASET DETAILS

**Kuhn Poker Agent Dataset**

In the Kuhn Poker task, we constructed a balanced and diverse dataset by generating match trajectories from three complementary opponent pairings: RandomAgent vs. GTOAgent to introduce non-strategic noise against equilibrium play, GTOAgent vs. GTOAgent to capture distributions close to Nash equilibrium, and BluffAgent vs. Counter-BluffAgent to model systematic over-bluffing and its corresponding counter-strategies. These matchups generated 32k, 64k, and 32k raw samples respectively, from which we selected a final 64k training set with an 8:2 train–validation split (51.2k : 12.8k). Importantly, GOE-LLM and the GTO-LLM baseline were both trained under identical hyperparameter settings, including a batch size of 512. Because checkpoints were saved every 100 steps, one full training run corresponds to processing 51.2k samples, ensuring that both models are evaluated after being exposed to exactly the same amount of data. GOE-LLM constructs a 64k training corpus combining diverse opponent styles, whereas the baseline GTO-LLM dataset (96k total samples) is exclusively generated from GTOAgent vs. GTOAgent interactions.

**Leduc Hold'em Agent Dataset**

For Leduc Hold'em, we followed a comparable data-generation protocol while expanding the behavioral coverage of strategic deviations. Specifically, the dataset combines samples from RandomAgent vs. GTOAgent (24k), GTOAgent vs. GTOAgent (32k), and TightAgent vs. LooseAgent (24k) matchups, jointly modeling equilibrium play, non-strategic noise, and classical tight–loose exploitation dynamics. From these raw 80k trajectories, we constructed an 80k training dataset with an 8:2 train–validation split (64k : 16k). This ensures that the training distribution does not inadvertently encode cyclical or anti-correlated behavioral patterns, but instead reflects a broad yet stable mixture of strategically interpretable interactions. Consistent with the Kuhn Poker setting, both GOE-LLM and the GTO-LLM baseline were trained with identical optimization hyperparameters, such that each reported checkpoint corresponds to the same effective exposure—64k samples—with the baseline GTO-LLM relying entirely on GTOAgent vs. GTOAgent equilibrium-only data during construction.

Table 4: Composition of training data for Kuhn Poker and Leduc Hold'em. The table reports the generated samples from each matchup and the final selected training size.

| Game | Matchup | Generated Samples | Training Samples(Train:Val) |
|---|---|---|---|
| Kuhn Poker | Random vs. GTO | 32,000 | 64,000(51,200:12,800) |
| | GTO vs. GTO | 64,000 | |
| | Bluff vs. Counter-Bluff | 32,000 | |
| Leduc Hold'em | Random vs. GTO | 24,000 | 80,000(64,000:16,000) |
| | GTO vs. GTO | 32,000 | |
| | Tight vs. Loose | 24,000 | |

## A.3 PROMPT DESIGN

---

**Prompt 1: Kuhn Poker Agent**

You are an expert Kuhn Poker player, tasked with making optimal decisions in a two-player simplified poker environment. Your objective is to maximize expected value while reasoning about hidden information and opponent behavior.

## Game Rules
Kuhn Poker is a fundamental imperfect-information game with the following properties:
- Deck: 3 cards {J, Q, K}, ranked J < Q < K.
- Initial setup: Each player antes 1 chip and is dealt 1 private card (cards are dealt without replacement).
- Showdown: If both players remain without folding, the player with the higher card wins the pot.

## Action Rules
At each decision point, the following actions may be taken (subject to the betting history):
- [Check]: Pass without betting (only if no bet is on the table).
- [Bet]: Add 1 chip to the pot (only if no bet is on the table).
- [Call]: Match an opponent's bet by adding 1 chip to the pot (only if a bet has been made).
- [Fold]: Concede the pot to the opponent (only if a bet has been made).

## Decision Context (State Representation)
The state will be described in natural language, including:
- Your role: Player {player_id} ({first} to act this round).
- Private information: Your card = {card}.
- Action history: {history}.
- Legal actions available: {available_actions}.

## Output Format
Your response must explicitly include reasoning and action selection:

```
<think> Your thoughts and reasoning </think>
<answer> [ACTION] </answer>
```

- The `<think>` field should describe your strategic reasoning (e.g., hand strength, bluffing, opponent modeling).
- The `<answer>` field must contain exactly one action from {available_actions}.

## Important Notes
1. You must always provide a reasoning process in the `<think>` field.
2. Your final choice in `<answer>` must be strictly one of the available actions.
3. Decisions should consider both exploitative opportunities and minimization of exploitability.

---

---

**Prompt 2: Leduc Hold'em Agent**

You are an expert Leduc Hold'em player, tasked with making optimal strategic decisions in a two-player fixed-limit poker environment. Your goal is to maximize expected utility through well-reasoned betting actions, given private information (your card), public information (community card, betting history), and the opponent's behavior.

## Game Rules
Leduc Hold'em is a simplified poker variant with perfect recall and incomplete information:
- Deck: 6 cards (two each of J, Q, K).
- Initial setup: Each player antes 1 chip and receives 1 private card.
- Round 1 (Pre-flop): A betting round begins. Bet size is 2 chips. Maximum of 2 raises per round.
- Round 2 (Post-flop): One public card is revealed. A second betting round begins with fixed bet size of 4 chips. Maximum of 2 raises per round.
- Showdown: If both players remain, the winner is determined by hand strength .
- Pot Limitations: Betting is fixed-limit, ensuring bounded strategy space.

## Action Rules
At each decision point, you may choose from the following legal actions (subject to game constraints):
- [Check]: Pass without adding chips (only if no outstanding bet).
- [Bet]: Initiate betting (2 chips pre-flop, 4 chips post-flop; only if no outstanding bet).
- [Raise]: Increase the current bet by the fixed size (2 or 4 chips), only if fewer than 2 raises have occurred this round.
- [Call]: Match the opponent's current bet (only if a bet exists).
- [Fold]: Concede the pot immediately (only if a bet exists).

## Decision Context (State Representation)
The current decision state is described in natural language and includes:
- Your role: Player {player_id} ({first} to act this round).
- Private information: Your hole card = {card}.
- Public information: {board_card}.
- Action history: {history}.
- Legal actions available: {available_actions}.

## Output Format
Your response must explicitly include both reasoning and action selection:

```
<think> Detailed reasoning about hand strength, betting history,
opponent modeling, and risk/reward tradeoffs. </think>
<answer> [ACTION] </answer>
```

- The `<think>` field should explain why you select a given action.
- The `<answer>` field must contain exactly one action from {available_actions}.

## Important Notes
1. Always provide reasoning in the `<think>` section before deciding.
2. Your decision should balance value extraction, bluffing opportunities, and minimization of exploitability.
3. The final output must conform strictly to the format above.
4. You must never output actions not listed in {available_actions}.

---

### A.4 MLP PROFILER DETAILS

The MLP Opponent Profiler is designed to enhance the generalization capability of the LLM Exploiter against unseen opponents. Specifically, we maintain a history of the opponent's behavior trees over the last $k$ games. We extract features from these decision trees to form a fixed-length vector representation. This vector is then fed into a pre-trained MLP classifier that maps it to a discrete opponent type space. Finally, we translate the identified opponent type into a language-based description, which is provided as auxiliary information to the LLM Exploiter. This enables the LLM to adjust its strategy based on the classified opponent type.

The value of $k$ is a critical hyperparameter that balances the trade-off between responsiveness and stability in opponent modeling. A smaller $k$ allows the profiler to quickly adapt to recent changes in the opponent's strategy, while a larger $k$ provides a more stable and comprehensive view of the opponent's behavior over time.

We visualize the decision feature vectors for different values of $k$ using t-SNE in Figure 6. In Kuhn Poker, when $k$ is small, the distribution of decision vectors is more scattered, making it difficult to distinguish between different opponent types. However, when $k = 10$, the decision vectors form distinct clusters, with clearer boundaries between different opponent types. This clustering effect facilitates the training and generalization of the MLP classifier. For the more complex Leduc Hold'em poker, which has a larger strategy space, a longer history $k$ is needed to capture the opponent's be-

havior patterns. We find that when $k = 50$, the decision vectors also form distinct clusters in the t-SNE visualization, aiding in distinguishing between different opponent types.

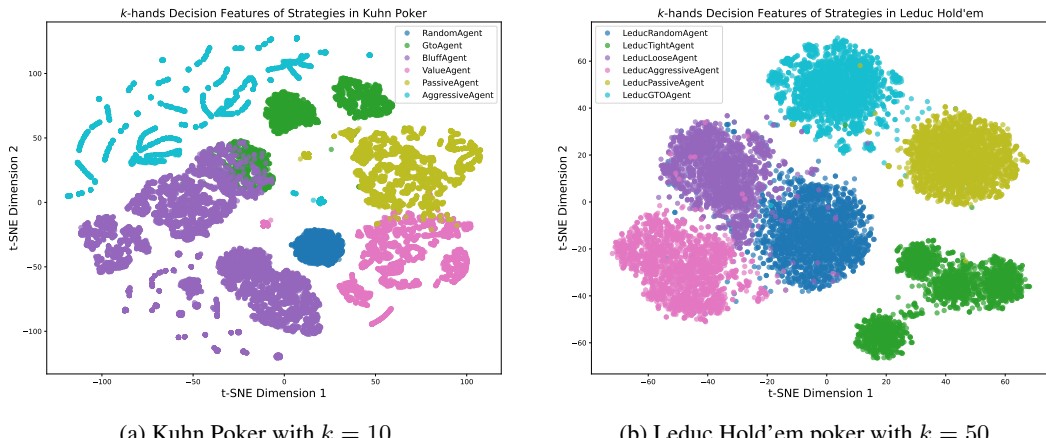

(a) Kuhn Poker with $k = 10$           (b) Leduc Hold'em poker with $k = 50$

Figure 6: More complex games require longer history $k$ to capture opponent behavior patterns.

Table 5: Confusion matrix across six strategy categories. Most classes show high classification accuracy, with Value(s), Passive(s), and Aggressive(s) achieving near-perfect performance.

| True \ Pred | Random | GTO(s) | Bluff(s) | Value(s) | Passive(s) | Aggressive(s) | Accuracy |
|---|---|---|---|---|---|---|---|
| Random | 600 | 0 | 0 | 0 | 0 | 0 | 1.0000 |
| GTO(s) | 0 | 1730 | 69 | 0 | 1 | 0 | 0.9611 |
| Bluff(s) | 0 | 61 | 4739 | 0 | 0 | 0 | 0.9873 |
| Value(s) | 0 | 0 | 0 | 2392 | 8 | 0 | 0.9967 |
| Passive(s) | 0 | 4 | 0 | 3 | 2393 | 0 | 0.9971 |
| Aggressive(s) | 0 | 0 | 0 | 0 | 0 | 2400 | 1.0000 |

## A.5 TRAINING PARAMETERS

**Opponent Profiler.**

Table 6: Training parameters for the Opponent Profiler.

| Parameter | Value |
|---|---|
| Hidden Layer Sizes | (128, 64) |
| Activation Function | ReLU |
| Solver | Adam |
| L2 Regularization Coefficient ($\alpha$) | 1e-4 |
| Batch Size | 64 |
| Learning Rate | 1e-3 |
| Maximum Iterations | 200 |
| Early Stopping | Enabled |
| Patience (No Change Tolerance) | 10 iterations |
| Validation Fraction | 0.1 |

**LLM Exploiter.**

Table 7: Training parameters employed in the LLM Exploiter experiment.

| Parameter | Value |
|---|---|
| Algorithm Advantage Estimator | grpo |
| Training Batch Size | 512 |
| Maximum Prompt Length | 1024 |
| Maximum Response Length | 512 |
| Overlong Prompt Filtering | False |
| Truncation Strategy | error |
| Learning Rate | 1e-6 |
| PPO Mini-batch Size | 128 |
| PPO Micro-batch Size per GPU | 16 |
| KL Divergence Loss | Enabled |
| KL Loss Coefficient | 0.001 |
| KL Loss Type | low_var_kl |
| Entropy Coefficient | 0 |
| Gradient Checkpointing | Enabled |
| KL in Reward Computation | Disabled |
| Critic Warmup Steps | 0 |
| Rollout Sample Size | 5 |

## A.6 TRAINING ENVIRONMENT DETAILS

Table 8: Integrated environment configuration for VERL training (hardware, software, memory, distributed training, inference, and data preprocessing).

| Category | Parameter | Value |
|---|---|---|
| Hardware | GPUs per Node | 2 |
| | Number of Nodes | 1 |
| | Tensor Model Parallel Size | 2 |
| | GPU Memory Utilization | 0.9 |
| Software | Training Framework | VERL |
| | Inference Engine | vLLM |
| | Logging System | Console + Weights & Biases |
| Memory | Gradient Checkpointing | Enabled |
| | Parameter Offload (Actor) | False |
| | Parameter Offload (Reference) | True |
| | Optimizer Offload | False |
| | Remove Padding | True |
| Distributed | FSDP Parameter Offload | False |
| | FSDP Optimizer Offload | False |
| | Micro-batch Size per GPU | 16 |
| | Log-Prob Micro-batch Size | 32 |
| | Multi-GPU Communication | NCCL |
| Inference | Inference Backend | vLLM |
| | Rollout Samples | 5 |
| | Memory Management | Dynamic |
| | Batch Processing | Parallel |
| Data Preprocessing | Data Loading | Parquet Reader |
| | Sequence Truncation | Error on Overflow |
| | Prompt Filtering | Disabled |
| | Batch Processing | Dynamic Batching |

# B  MORE RESULTS

## B.1  ADDITIONAL RESULTS ON WIN RATE

Table 9: Win Rate of GOE-LLM with baseline methods in Leduc Hold'em. Each row represents an agent, each column represents a specific opponent strategy.

| LLM(3B) | | Opponent (as $P_{opp}$) | | | | | | |
|---|---|---|---|---|---|---|---|---|
| Method | Role | Random | GTO | Tight | Loose | Passive | Aggressive | Average |
| Random | P0 | 39.83 % | 38.67 % | 66.00 % | 26.67 % | 44.55 % | 20.61 % | 39.39 % |
| | P1 | 52.33 % | 41.00 % | 67.39 % | 31.89 % | 48.39 % | 28.94 % | 44.99 % |
| GTO | P0 | 49.67 % | 41.67 % | 53.61 % | 37.39 % | 44.83 % | 36.28 % | 43.91 % |
| | P1 | 43.00 % | 40.50 % | 53.00 % | 36.83 % | 41.22 % | 35.28 % | 41.64 % |
| BR | P0 | 48.42 % | 41.33 % | 58.35 % | 35.94 % | 45.28 % | 33.65 % | 43.83 % |
| | P1 | 43.83 % | 40.88 % | 56.75 % | 34.22 % | 44.83 % | 32.57 % | 42.18 % |
| LLM | P0 | 51.25 % | 59.17 % | 77.72 % | 11.61 % | 50.50 % | 21.11 % | 45.23 % |
| | P1 | 37.50 % | 37.08 % | 74.00 % | 20.17 % | 51.22 % | 25.83 % | 40.97 % |
| LLM+OM | P0 | 54.67 % | 35.67 % | 77.72 % | 26.67 % | 51.50 % | 11.78 % | 43.00 % |
| | P1 | 37.00 % | 34.67 % | 74.00 % | 31.89 % | 51.00 % | 20.17 % | 41.45 % |
| GOE-LLM | P0 | 43.50 % | 40.17 % | 40.78 % | 37.39 % | 40.33 % | 41.72 % | 40.65 % |
| | P1 | 41.17 % | 39.17 % | 42.72 % | 36.83 % | 38.83 % | 39.06 % | 39.63 % |

Table 10: Win Rate of GOE-LLM with baseline methods in Kuhn Poker.

| LLM(3B) | | Opponent (as $P_{opp}$) | | | | | | |
|---|---|---|---|---|---|---|---|---|
| Method | Role | Random | GTO | Bluff | Value | Passive | Aggressive | Average |
| Random | P0 | 56.67 % | 52.67 % | 49.07 % | 47.36 % | 55.13 % | 50.94 % | 51.97 % |
| | P1 | 44.17 % | 51.48 % | 37.57 % | 52.02 % | 57.09 % | 42.50 % | 47.47 % |
| GTO | P0 | 51.83 % | 52.08 % | 50.64 % | 50.59 % | 54.07 % | 51.10 % | 51.72 % |
| | P1 | 47.33 % | 51.59 % | 41.80 % | 51.20 % | 55.08 % | 45.53 % | 48.76 % |
| BR | P0 | 53.00 % | 51.82 % | 51.05 % | 50.00 % | 62.45 % | 50.00 % | 53.05 % |
| | P1 | 47.97 % | 50.78 % | 50.00 % | 50.00 % | 64.60 % | 50.00 % | 52.22 % |
| LLM | P0 | 64.30 % | 55.38 % | 54.51 % | 50.00 % | 56.01 % | 58.32 % | 56.42 % |
| | P1 | 50.10 % | 53.61 % | 41.15 % | 52.58 % | 60.57 % | 52.43 % | 51.74 % |
| LLM+OM | P0 | 60.33 % | 54.73 % | 54.02 % | 40.82 % | 61.95 % | 41.62 % | 52.24 % |
| | P1 | 44.30 % | 51.21 % | 39.30 % | 48.33 % | 64.30 % | 43.39 % | 48.47 % |
| GTO-LLM | P0 | 42.40 % | 44.45 % | 40.43 % | 49.99 % | 50.00 % | 37.92 % | 44.20 % |
| | P1 | 41.33 % | 47.17 % | 37.01 % | 50.00 % | 50.00 % | 34.84 % | 43.39 % |
| GOE-LLM | P0 | 48.77 % | 49.65 % | 49.30 % | 49.94 % | 61.22 % | 49.56 % | 51.41 % |
| | P1 | 48.20 % | 49.86 % | 48.88 % | 49.98 % | 61.98 % | 49.03 % | 51.32 % |

# C  ABLATION DETAILS

## C.1  MLP PROFILER.

As shown in Figure 7, we can see that GOE-LLM(w/o Profiler) and GOE-LLM perform similarly against seen opponents (e.g., Random, GTO, Bluff), but show clear differences against unseen opponents (Value, Passive, Aggressive). With the MLP Profiler, GOE-LLM generally achieves higher win rates against these unseen opponents, with the most notable improvements observed when P0 plays against Value and Passive opponents. On average, GOE-LLM enhances adaptability to unseen opponents while maintaining its performance against seen ones, demonstrating that the MLP Profiler effectively improves the model's generalization and robustness.

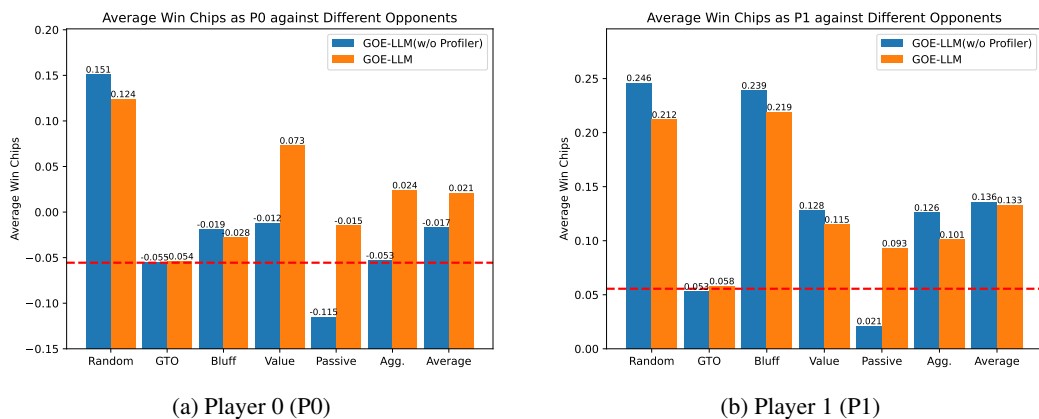

(a) Player 0 (P0)              (b) Player 1 (P1)

Figure 7: Critic score curves for LLM exploiters trained with and without the MLP-based opponent profiler, across different player positions (P0 and P1).

## C.2 MIXTURE OF TRAINING DATA.

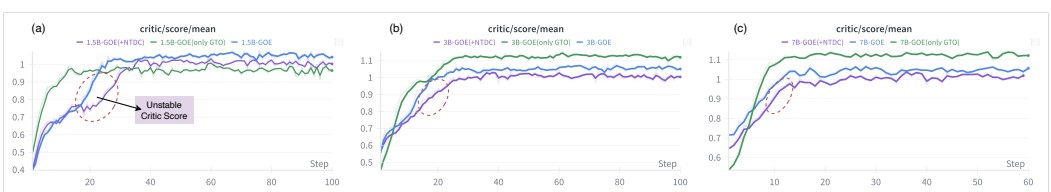

Figure 8: Critic score curves for LLM exploiters trained on different mixture datasets, across different model sizes (1.5B, 3B, 7B).

As shown in Figure 8, the critic score curves across different model sizes (1.5B, 3B, 7B) consistently demonstrate clear differences among the three mixture strategies. For the smallest model (1.5B), GOE-LLM achieves the balance between stability and performance, with faster improvement and higher final scores compared to GOE-LLM(only GTO), while GOE-LLM(+NTDC) suffers from severe fluctuations and degraded performance due to the instability introduced by non-transitive dominance cycles. At the 3B scale, GOE-LLM remains the most effective, outperforming the other two baselines; although the negative impact of NTDC is partially mitigated relative to 1.5B, the corresponding curve still exhibits noticeable instability. At the largest scale (7B), GOE-LLM continues to deliver the highest and most stable critic scores, but the performance gap with GOE-LLM(+NTDC) narrows, suggesting that larger models have stronger robustness to complex, cyclic opponent data. In contrast, only GTO remains consistently inferior across all model sizes, highlighting the importance of diversified yet principled data mixtures for effective exploitation learning.

