# OpenReview forum: "Generalizable Opponent Exploitation in LLM Agents via Mixed Best-Responses Training"
_ICLR.cc/2026/Conference — Submitted to ICLR 2026_

### Official Review · Reviewer_RXu2 · 2025-10-18

**Soundness:** 2
**Presentation:** 2
**Contribution:** 2
**Rating:** 2
**Confidence:** 4

**Summary:**

his paper introduces GOE-LLM (Generalizable Opponent Exploitation with LLMs), a framework aimed at enabling Large Language Model (LLM)-based agents to exploit weaknesses in opponents' strategies within two-player zero-sum imperfect-information games like poker. The approach features a two-tier architecture: (1) a lightweight Multi-Layer Perceptron (MLP) profiler that analyzes recent opponent behaviors to classify strategic patterns and convert them into natural language descriptions, and (2) an LLM exploiter fine-tuned via Group Relative Policy Optimization (GRPO) on a carefully curated mixture of best-response strategies. To promote training stability and generalization, the authors propose a "Mixture-of-Best-Responses" principle that avoids non-transitive dominance cycles in the training data. Evaluations on Kuhn Poker and Leduc Hold'em demonstrate GOE-LLM's effectiveness.

**Strengths:**

GOE-LLM's hybrid design—merging an MLP profiler for dynamic opponent modeling with an LLM exploiter—offers a practical, interpretable method to infuse behavioral insights into LLM decision-making, drawing inspiration from game-theoretic concepts like mixture-of-opponents learning.

The "Mixture-of-Best-Responses" principle stands out as a key innovation, providing a formal guideline to balance diversity in training data.

**Weaknesses:**

1. **Omission of Key Related Works**: The paper overlooks relevant literature on LLMs in repeated games, such as Akata et al. (2025) (Playing repeated games with large language models), which explores LLMs' strategic behavior in repeated interactions (e.g., Prisoner's Dilemma variants) and highlights their ability to adapt and exploit over iterations.

2. **Lack of Clarity on Seen vs. Unseen Opponents**: The distinction between "seen" (in-training) opponents and "unseen" (OOD) ones is not explicitly detailed in the main text, relying on Table 1 for inference. A clearer exposition—perhaps with examples of how the MLP profiler bridges this gap or visualizations of strategy spaces—would improve readability and highlight the generalization claims.

3. **Potential Overfitting to Synthetic Data**: All training data derives from simulated matchups, raising concerns about overfitting to these artificial patterns despite the mixture principle. It remains unclear how the framework would perform with real-game traces or human-player data, which introduce noise, variability, and evolving strategies not captured in rule-based simulations.

4. **Unclear Mechanism for Avoiding Overfitting via Mixture Principle**: While the Mixture-of-Best-Responses principle is posited to enhance generalization by excluding cyclic counters, the paper does not deeply explain *why* this specifically mitigates overfitting—e.g., through theoretical analysis of strategy diversity or convergence guarantees. Ablations show empirical benefits, but a more formal justification  would clarify its role beyond stability.

5. **Limited Exploration of Larger LLMs and Advanced Baselines**: Evaluations cap at 7B-parameter models, omitting frontier LLMs (e.g., GPT-4 scale), where in-context learning might suffice for opponent exploitation without fine-tuning. Baselines like LLM+OM (vanilla prompting with opponent info) lag behind, but incorporating stronger techniques—such as chain-of-thought prompting or self-reflection from works like Yao et al. (2023b)—could offer a more equitable comparison and assess if GOE-LLM's gains persist at scale.

6. **Questionable Overall Contribution Given Win Rate Metrics**: Although GOE-LLM excels in expected value (chips per hand), its average win rates (Tables 8 and 9) are lower than several baselines (e.g., LLM, GTO, BR) across opponents and games shown in Tables 8 and 9 (see the last column Average).  These results make the contributions weak.

Minor:

Fonts size in Figure 3 is too small.

**Questions:**

What is the role of Figures 3 and 4? Both figures were never mentioned.

---

> ### Author Response · Authors · 2025-11-22
> **Response to Reviewer RXu2 (1/3)**
>
> We appreciate your comments. Our detailed answers are provided below.
>
> ---
> ### **Response to Weakness 1:**
>
> *Q: Omission of Key Related Works*
>
> We appreciate the reviewer for pointing out the related work by Akata et al. (2025). While this work indeed explores LLMs in repeated games, our focus is on the specific challenge of opponent exploitation through fine-tuning LLMs. We have added a discussion of this related work in the revised manuscript to clarify the distinctions and contributions of our approach.
>
> ---
> ### **Response to Weakness 2:**
>
> *Q: Lack of Clarity on Seen vs. Unseen Opponents*
>
> Thanks for the feedback.
> We provide a detailed description of the MLP profiler in **Appendix A.4** and include a t-SNE visualization (**Figure 6**) that illustrates the distributional differences between strategies of seen and unseen opponents. This helps clarify how the profiler connects in-training opponents with OOD ones by embedding their behaviors into a shared strategy space. In the revision, we have added an explicit explanation of this distinction in the main text to improve readability and more clearly highlight our generalization claims.
>
> ---
>
> ### **Response to Weakness 3:**
>
> *Q: Potential Overfitting to Synthetic Data*
>
> We acknowledge the concern regarding potential overfitting to synthetic data. The training data in our framework is generated through interactions among diverse AI opponents within the Poker environment, rather than artificially crafted patterns; **Appendix A.1 provides detailed information on this process.**
>
> We also fully support the use of real human-game traces. However, collecting such data involves substantial cost and introduces large variance, placing it beyond the scope of this work. We consider this an important direction for future research.

---

> ### Author Response · Authors · 2025-11-22
> **Response to Reviewer RXu2 (2/3)**
>
> ---
> ### **Response to Weakness 4:**
>
> *Q: Unclear Mechanism for Avoiding Overfitting via Mixture Principle*
>
> Overfitting is indeed a complex issue. We have provided extensive empirical evidence through ablation studies and loss visualizations (**Figure 3**) to demonstrate the stability and convergence of our Mixture-of-Best-Responses approach. While theoretical guarantees are challenging in this context, we will clarify the empirical rationale behind our method in the revised manuscript.
>
> ---
> ### **Response to Weakness 5:**
>
> *Q: Limited Exploration of Larger LLMs and Advanced Baselines*
>
> We appreciate the reviewer's suggestion to evaluate larger LLMs. In response, we have conducted additional experiments using GPT-4o-mini, a more advanced model, to assess its opponent exploitation capabilities both with and without our MLP Profiler. However, we believe that incorporating more advanced prompting techniques (e.g., chain-of-thought prompting or self-reflection) is beyond the scope of our current work, which focuses on opponent exploitation through fine-tuning rather than prompt engineering.
>
> All methods were tested with the same prompt template as used in GOE.
> To ensure statistical reliability, each model played 3,000 hands against each of 24 opponents (from 6 types), resulting in 72K total hands. Same as in the main experiments.
>
> The results are shown below, **Bold values highlight cases that fall below equilibrium expectations(−1/18 as P0, +1/18 as P1)**:
>
> | Method          | Role | Random  | GTO(s)      | Bluff(s)    | Value(s)    | Passive(s)  | Aggressive(s) | Average  |
> |-----------------|------|---------|----------|----------|----------|----------|------------|----------|
> | GPT-4o-mini     | P0   | 0.3516  | **-0.1096**  | **-0.1112**  | -0.0504  | **-0.1165**  | **-0.1468**    | -0.0305  |
> |                 | P1   | 0.3120  | **-0.0241**  | 0.1360   | **-0.0997**  | **-0.1200**  | 0.1348     | 0.0565   |
> | GTP-4o-mini+MLP Profiler | P0   | 0.4403  | **-0.1105**  | **-0.1217**  | 0.1376   | 0.0714   | 0.0687     | 0.0809   |
> |                 | P1   | 0.4073  | **-0.0336**  | 0.1517   | 0.1221   | 0.1029   | 0.1156     | 0.1443   |
> | GOE-3B         | P0   | 0.1240  | -0.0543  | -0.0283  | 0.0733   | -0.0148  | 0.0235     | 0.0206   |
> |                 | P1   | 0.2120  | 0.0577   | 0.2185   | 0.1148   | 0.0928   | 0.1013     | 0.1328   |
>
> Key findings:
> - GPT-4o-mini does not reliably exploit opponents via ICL alone.
> - Augmenting GPT-4o-mini with our MLP Profiler significantly enhances its exploitation capabilities.
> - However, even with this enhancement, GPT-4o-mini remains less stable than GOE, particularly when acting as P0 or facing the GTO opponent. In contrast, GOE achieves stable above-equilibrium performance across all opponent types.
>
> These results support our claim that:
> 1. **Strong LLMs alone do not reliably perform opponent exploitation through ICL, and a dedicated opponent modeling mechanism is essential.**
> 2. **GOE provides a more stable and generalizable solution than simply prompting a large LLM.**
>
> Meanwhile, we also conducted experiments using GPT-5.1, playing 600 hands against each of 6 opponents (5 extreme different style opponents and 1 NE strategy), resulting in a total of 3600 hands. The results are as follows.
>
> | Method           | Role | Random | GTO(1/3) | Bluff(1,1) | Value(1) | Passive(1) | Aggressive(1) | Average |
> |------------------|------|--------|----------|------------|----------|------------|---------------|---------|
> | GPT-5.1+MLP Profiler     | P0   | 0.183  | **-0.077**   | -0.065     | 0.078    | 0.037      | **-0.162**        | -0.001  |
> |                  | P1   | 0.227  | 0.068    | 0.282      | **-0.058**   | 0.083      | 0.078         | 0.113   |
> | GOE-3B          | P0   | 0.124  | -0.059   | 0.019      | 0.086    | 0.049      | 0.091         | 0.052   |
> |                  | P1   | 0.212  | 0.062    | 0.297      | 0.154    | 0.113      | 0.135         | 0.162   |
>
> It can be observed that even with a powerful model, there are still challenges in surpassing equilibrium returns when facing extreme style opponents.

---

> ### Author Response · Authors · 2025-11-22
> **Response to Reviewer RXu2 (3/3)**
>
> ---
> ### **Response to Weakness 6:**
>
> *Q: Questionable Overall Contribution Given Win Rate Metrics*
>
> We appreciate the reviewer's observation regarding win rate metrics. **However, in poker, the primary objective is to maximize expected value (chips won per hand) rather than win rate alone**. A strategy that maximizes expected value may involve taking calculated risks that do not necessarily translate to a higher win rate. We have clarified this distinction in the revised manuscript to better explain the significance of our results in the context of poker strategy.
>
> ---
> ### **Response to Question 1 and Minor:**
>
> *Q: What is the role of Figures 3 and 4? Small font size in Figure 3.*
>
> Thank you for pointing out the small font size and incorrect citation; we have corrected it in the revision.
>
> Figure 3 visualizes the impact of different data mixture strategies on model performance across various LLM sizes, highlighting the effectiveness of our Mixture-of-Best-Responses approach. Figure 4 compares the average win chips per hand for GOE-LLM with and without the MLP Profiler, demonstrating the significant contribution of the profiler to opponent exploitation capabilities. We have ensured that these figures are now properly referenced and explained in the revised manuscript.
>
> ---
>
> We hope these responses adequately address your concerns. Please let us know if further clarification is needed.

---

> > ### Comment · Reviewer_RXu2 · 2025-11-26
> >
> > Thanks for your response. I still think the win rate result is unreasonable, and advanced baselines should be included.

---

> > > ### Author Response · Authors · 2025-11-27
> > > **Response to Reviewer RXu2**
> > >
> > > ---
> > >
> > > Thank you for your follow-up comments.
> > >
> > > ### **Response to Advanced Baselines:**
> > >
> > > As addressed in our previous response, we have already included experiments with GPT-4o-mini and GPT-5.1, both with and without our MLP Profiler. The findings show:
> > >
> > > - Even very strong LLMs cannot reliably perform opponent exploitation through in-context learning alone.
> > >
> > > - Adding our MLP Profiler improves exploitation ability, but the results remain less stable and less consistent than GOE-LLM.
> > >
> > > - GOE-LLM produces robust above-equilibrium performance across all opponent types.
> > >
> > > These observations reinforce the necessity of the GOE framework and demonstrate that simply using a larger LLM or more advanced prompting is insufficient for reliable opponent exploitation.
> > >
> > > ---
> > > ### **Response to Win Rate Results:**
> > >
> > > As shown in Tables 9 and 10 of the revised manuscript, while GOE-LLM exhibits a slightly lower win rate, it achieves a substantial improvement in expected value (chips won per hand). To further clarify why this phenomenon is reasonable, we conducted additional large-scale experiments (24 opponents × 6,000 hands), and collected richer statistics.
> > >
> > > In our setting, each hand results in one of two pot sizes, 2 or 4 chips. These correspond to the following outcomes:
> > >
> > > - Pot 2: win +2 or lose –2
> > > - Pot 4: win +4 or lose –4
> > >
> > > Let the number of outcomes be:
> > > - +2: a, −2: b --> net count for Pot 2 is x = a − b
> > > - +4: c, −4: d --> net count for Pot 4 is y = c − d
> > >
> > > **A larger x means winning more small pots; a larger y means winning more big pots.**
> > >
> > > The detailed results are as follows. To emphasize the impact of large-pot outcomes, **the bold entries in the tables mark the largest y value for each opponent column**.
> > >
> > > | Methods(as P0) | Random   |          | Gto(s)      |          | Bluff(s)    |          | Value(s)    |          | Passive(s)  |          | Aggressive(s) |          |  Average |          |
> > > |----------------|----------|----------|----------|----------|----------|----------|----------|----------|----------|----------|------------|----------|----------|----------|
> > > |                | x(Pot 2) | y(Pot 4) | x(Pot 2) | y(Pot 4) | x(Pot 2) | y(Pot 4) | x(Pot 2) | y(Pot 4) | x(Pot 2) | y(Pot 4) | x(Pot 2)   | y(Pot 4) | x(Pot 2) | y(Pot 4) |
> > > | LLM            | 765      | 93       | 1083     | -760     | 1004     | -733     | 748      | -748     | 1165     | -805     | 1241       | -742     | 1001     | -616     |
> > > | LLM+OM         | 574      | 46       | 1048     | -764     | 1015     | -774     | -538     | -14      | 1691     | -974     | -369       | -134     | 570      | -436     |
> > > | GOE            | -519     | **445**      | 122      | **-143**     | 1        | **-43**      | -226     | **223**      | 1391     | **-718**     | -123       | **97**       | 108      | **-23**      |
> > >
> > >
> > > | Methods(as P1) | Random   |          | GTO(s)      |          | Bluff(s)    |          | Value(s)    |          | Passive(s)  |          | Aggressive(s) |          |  Average |          |
> > > |----------------|----------|----------|----------|----------|----------|----------|----------|----------|----------|----------|------------|----------|----------|----------|
> > > |                | x(Pot 2) | y(Pot 4) | x(Pot 2) | y(Pot 4) | x(Pot 2) | y(Pot 4) | x(Pot 2) | y(Pot 4) | x(Pot 2) | y(Pot 4) | x(Pot 2)   | y(Pot 4) | x(Pot 2) | y(Pot 4) |
> > > | LLM            | -149     | 155      | 761      | -544     | -564     | 32       | 943      | -788     | 1456     | -822     | 110        | 36       | 426      | -322     |
> > > | LLM+OM         | -430     | 88       | 575      | -502     | -628     | -14      | 71       | -171     | 1821     | -963     | -456       | 60       | 159      | -250     |
> > > | GOE            | -852     | **744**      | -191     | **182**      | -791     | **724**      | -346     | **345**      | 1160     | **-441**     | -421       | **363**      | -240     | **319**      |
> > >
> > > Key Observations:
> > >
> > > - **GOE-LLM obtains dramatically higher net wins in Pot-4 situations (y), far exceeding other methods.** This directly contributes to its much higher expected value(EV), since Pot-4 outcomes dominate chip returns.
> > >
> > > - GOE-LLM sometimes sacrifices Pot-2 win rate (x), which leads to a smaller overall win rate. However, this is strategically rational: **the EV impact of Pot-4 is twice that of Pot-2, and maximizing long-term chip gain requires prioritizing high-value opportunities**.
> > >
> > > - This pattern reflects a well-known phenomenon in poker strategy: **maximizing EV is not equivalent to maximizing win rate**, and optimal exploitative play often trades small-pot frequency for large-pot profitability.
> > >
> > > These results provide a clear explanation for the seemingly counterintuitive win-rate behavior in Tables 9 and 10.
> > >
> > > Thank you again for your constructive feedback. We hope these additional analyses and experiments clearly address your concerns.

---

### Official Review · Reviewer_Jxaf · 2025-10-31

**Soundness:** 3
**Presentation:** 3
**Contribution:** 3
**Rating:** 6
**Confidence:** 3

**Summary:**

This paper proposes GOE-LLM, a novel framework for training LLM agents to generalize opponent exploitation in two-player zero-sum imperfect-information games. Unlike existing LLM-based agents that focus on equilibrium play or prompt engineering, GOE-LLM explicitly targets the ability to exploit opponents. The framework consists of a MLP-based opponent profiler and an LLM exploiter, which is fine-tuned using a Mixture-of-Best-Responses training principle. The policy is optimized using a modified Group Relative Policy Optimization (GRPO) algorithm with an opponent-aware reward structure. Extensive experiments in Kuhn and Leduc Poker show that GOE-LLM outperforms strong baselines across seen and unseen opponents, demonstrating robust exploitation capabilities and effective generalization.

**Strengths:**

1. This paper aims to solve a novel problem, i.e., generalizable opponent exploitation for LLM agents in imperfect-information games, which is largely unaddressed in prior work.
2. The work is methodologically sound, with well-motivated design choices and rigorous evaluation. The integration of GRPO with a fine-grained, opponent-aware reward structure is technically strong.
3. The work makes a significant contribution to both the LLM and multi-agent learning communities. It demonstrates how LLM agents can move beyond equilibrium play and adapt to exploit suboptimal opponents, which is crucial for strategic reasoning tasks in real-world applications.

**Weaknesses:**

1. The paper evaluates GOE-LLM exclusively on Kuhn Poker and Leduc Hold’em, which are small-scale, two-player zero-sum games with limited action and state spaces. While these environments are standard testbeds, they do not capture the complexity of real-world strategic interactions or large-scale imperfect-information games.
2. The MLP Profiler is a key component in the framework, yet the paper does not deeply analyze its accuracy or how profiling errors affect downstream exploitation. For example, misclassifying a bluffing agent as passive might lead to poor decisions by the LLM exploiter.
3. The training of the profiler assumes access to clean, labeled data from known opponent types. However, in many realistic settings, agents must learn to exploit opponents without such strong supervision.

**Questions:**

1. The framework assumes the profiler can reliably map short-term behavioral traces (e.g., past 10 or 50 games) to meaningful strategy categories. Could the authors share accuracy metrics of the profiler on seen vs. unseen opponents, and discuss how misclassification impacts downstream performance?
2. The paper defines a non-transitive dominance cycle and mentions selecting mixtures to avoid it. Is this done manually, heuristically, or through some automated graph-based filtering process?
3. In Table 2, why do some agents outperform BR*? For example, LLM achieves 0.317 (P0) and GOE-LLM 0.212 (P1) against the Random opponent, while BR* only gets 0.175 (P0) and 0.181 (P1). Could the authors explain this discrepancy?

---

> ### Author Response · Authors · 2025-11-22
> **Response to Reviewer Jxaf (1/2)**
>
> We sincerely thank the reviewer for the thoughtful and constructive feedback.
>
> ---
> ### **Response to Question 1 & Weaknesses 2**
>
> *Q: MLP Profiler accuracy and impact of misclassification*
>
> Thank you for highlighting this important point. We computed the classification accuracy of the MLP profiler, which maps a 24-dimensional decision-tree feature vector to one of the 6 opponent strategy categories.
> **The hidden layer sizes of the MLP are (128, 64), and the confusion matrix after training is as follows:**
>
> | True \ Pred     | Random | GTO(s) | Bluff(s) | Value(s) | Passive(s) | Aggressive(s) |
> |-----------------|-------------|----------|------------|------------|--------------|-----------------|
> | Random     | **600**         | 0        | 0          | 0          | 0            | 0               |
> | GTO(s)        | 0           | **1730**     | 69         | 0          | 1            | 0               |
> | Bluff(s)      | 0           | 61       | **4739**       | 0          | 0            | 0               |
> | Value(s)      | 0           | 0        | 0          | **2392**       | 8            | 0               |
> | Passive(s)    | 0           | 4        | 0          | 3          | **2393**         | 0               |
> | Aggressive(s) | 0           | 0        | 0          | 0          | 0            | **2400**            |
>
> The results demonstrate that the profiler generalizes well across all categories, with only minimal confusion between GTO and Bluff/Passive/Value.
> **Each opponent type achieves over 96% classification accuracy.**
>
> That indicates that the MLP Profiler has sufficient model capacity to effectively address the opponent classification task, thereby ensuring the accuracy of the provided opponent modeling information.
> In the revised version, we discuss this MLP profiler accuracy and its implications for downstream exploitation performance in ‘Contribution of MLP Profiler’ of Section 4.3.

---

> ### Author Response · Authors · 2025-11-22
> **Response to Reviewer Jxaf (2/2)**
>
> ---
> ### **Response to Question 2:**
>
> *Q: Handling non-transitive dominance cycles*
>
> We appreciate the reviewer’s question regarding how to avoid cyclic counter relations.
> Specifically, we **heuristically** select a subset of opponent strategies for generating training data to ensure diversity while avoiding cyclic dominance.
> As shown in Figure 2 of the main paper, Bluffing, Passive, and ValueBet form a non-transitive dominance cycle. However, by heuristically selecting only GTO, Bluffing, and Random for mixture training, we effectively avoid such cyclic relations.
>
> We further conducted 30k repeated matchups to validate the transitivity among our selected strategies:
> - GTO vs. Bluffing: as_player_0 = −0.0540, as_player_1 = 0.0636; −0.0540 + 0.0636 > 0 → **GTO > Bluffing**
>
> - GTO vs. Random: as_player_0 = 0.1663, as_player_1 = 0.1694; 0.1663 + 0.1694 > 0 → **GTO > Random**
>
> - Bluffing vs. Random: as_player_0 = 0.2204, as_player_1 = 0.1677; 0.2204 + 0.1677 > 0 → **Bluffing > Random**
>
> Thus, the final hierarchy is: **GTO > Bluffing > Random**.
>
> By heuristically choosing this transitive set of strategies, we effectively avoid non-transitive dominance cycles that could destabilize training, visulized in Figure 3 of the main paper. This design improves both the stability of the training process and the final performance of our model.
>
> ---
> ### **Response to Question 3:**
>
> *Q: Why some agents outperform BR in Table 2*
>
> Thank you for noting this discrepancy. After re-examining the best-response (BR) strategies, we found that the BR strategy against the Random opponent is not the Nash equilibrium policy but instead a significantly more aggressive policy that exploits Random’s unbalanced betting patterns. We validated this finding using 30,000 hands of simulation. The updated results are shown below:
>
> | Method          | Role | Random | GTO     | Bluff   | Value   | Passive | Aggressive | Average |
> |-----------------|------|--------|---------|---------|---------|---------|------------|---------|
> | BR              | P0   | **0.5088** | -0.0633 | 0.0951  | 0.1465  | 0.0810  | 0.0840     | **0.1420**  |
> |                 | P1   | **0.4145** | 0.0520  | 0.2431  | 0.1255  | 0.1253  | 0.1268     | **0.1812**  |
>
> We have updated the corresponding values (highlighted in bold above) in the revised manuscript. This correction explains why some LLM-based agents appeared to exceed BR*: the earlier BR* numbers underestimated the true exploitability of the Random opponent.
>
> ---
> Thanks again for the valuable feedback. We believe these clarifications and corrections significantly enhance the quality of our work.

---

> > ### Comment · Reviewer_Jxaf · 2025-11-26
> >
> > Thanks for the detailed response! The responses have answered my questions, and thus I maintain my original score.

---

### Official Review · Reviewer_BhQn · 2025-10-31

**Soundness:** 3
**Presentation:** 3
**Contribution:** 3
**Rating:** 6
**Confidence:** 2

**Summary:**

This paper proposes GOE-LLM, a framework that leverages LLM agents in two-player zero-sum games: an MLP is used to analyze opponent behavior, mixed best-response training is applied to learn opponent-exploitation strategies, and the goal is achieved via fine-tuning the LLM together with prompting. The method is validated on two-player zero-sum games such as Kuhn Poker and Leduc Hold’em, improving the agent’s ability to adapt to and exploit opponents who are suboptimal or deviate from equilibrium play.

**Strengths:**

- The paper raises a new question: how to perform opponent exploitation with LLMs. Traditional work in game theory and reinforcement learning typically focuses on learning equilibrium strategies or adapting to fixed opponents, whereas this paper seeks to enable LLMs to adapt to and effectively exploit diverse opponent weaknesses, especially those deviating from optimal strategies. This problem statement is innovative.

- The paper evaluates the framework on classic two-player zero-sum poker games—Kuhn Poker and Leduc Hold’em—as testbeds. Through comparisons with multiple baselines, it clearly demonstrates GOE-LLM’s advantages against various opponent strategies. The empirical results and data strongly support the method’s effectiveness.

- The paper carefully explains the design and implementation of the GOE-LLM framework—from opponent modeling to the exploiter’s training process. In particular, the mixed best-response training principle and the opponent-aware reward mechanism are described clearly, making the framework’s working principles easy to follow.

**Weaknesses:**

- Although the MLP Profiler achieves some success, its training data are primarily built around a small set of canonical opponent strategies (e.g., Random, GTO, Bluff). This may limit performance against highly complex or atypical opponents, especially in dynamic real-world settings. Enhancing generalization to unseen, unconventional strategies remains challenging.

- The validation focuses on Kuhn Poker and Leduc Hold’em. While widely used in game-theoretic research, these environments are relatively simple compared to real competitive settings (e.g., real poker tournaments or multi-player games). For practical deployment, more complex environments may be needed to test GOE-LLM’s generality and utility.

- The paper notes performance differences across LLM sizes and improvements with larger models, but the analysis is brief and does not delve into underlying causes (e.g., model capacity vs. data diversity vs. strategy generalization).

**Questions:**

- While the mixed best-response principle helps stabilize training, does mixing strategies against highly diverse opponents induce non-transitive dominance issues? The paper mentions avoidance but does not present detailed stability analyses.

- The focus is on zero-sum games, but real scenarios often involve multi-agent interaction, cooperation, and non-zero-sum payoffs. In such cases, pure “opponent exploitation” could over-optimize adversarial behavior and lead to instability.

- Although mixed best-response training is used, the construction of the mixture and the dataset effects are not fully detailed. How is diversity ensured, and how do you avoid over-reliance on certain strategy types?

---

> ### Author Response · Authors · 2025-11-22
> **Response to Reviewer BhQn (1/3)**
>
> Thank you for your thorough review and positive feedback on our work. Below, we provide detailed responses to each of your points.
>
> ---
>
> ### **Response to Weakness 1**
>
> *Q: Generalization to Unseen or Unconventional Opponents*
>
> Thanks for raising this important point.
> Our experimental environments include Kuhn Poker and Leduc Hold’em, both of which are multi-round, imperfect-information games with nontrivial complexity. In future work, we plan to evaluate GOE-LLM in more complex game environments to further assess its generalization ability.
>
> In addition, when training the LLM Exploiter, the training set only includes opponents from three strategies, while the remaining three opponent types (21 strategies) are completely unseen, as shown in **Table 1** of the main paper. To better illustrate this, we additionally report GOE’s performance on both seen and unseen tasks:
>
> | Methods         | Role | seen    | unseen  | average |
> |-----------------|------|---------|---------|---------|
> | Qwen2.5-3B      | P0   | 0.006   | -0.160  | -0.077  |
> |                 | P1   | -0.074  | -0.071  | -0.073  |
> | GOE(w/o MLP)-3B | P0   | 0.026   | -0.060  | -0.017  |
> |                 | P1   | 0.179   | 0.092   | 0.136   |
> | GOE-3B          | P0   | 0.014   | 0.027   | 0.021   |
> |                 | P1   | 0.163   | 0.103   | 0.133   |
>
> The LLM Exploiter fine-tuned with our method improves performance on both seen and unseen tasks.
> **For unseen opponents**, the base model Qwen2.5-3B performs weakly (P0: −0.160, P1: −0.071). GOE(w/o MLP)-3B improves this to (P0: −0.060, P1: 0.092), approaching the equilibrium payoff (−1/18) as P0 and exceeding +1/18 as P1. **With the addition of the MLP Profiler, GOE-3B further improves to (P0: 0.027, P1: 0.103), surpassing the equilibrium payoff as P0 and achieving a larger gain as P1.**
>
> These results demonstrate that our method effectively enhances performance even against unseen opponents, indicating robust generalization beyond the training distribution.

---

> ### Author Response · Authors · 2025-11-22
> **Response to Reviewer BhQn (2/3)**
>
> ---
>
> ### **Response to Weakness 2**
>
> *Q: Validation in More Complex Environments*
>
> We appreciate the suggestion and also acknowledge the limitations of Kuhn Poker and Leduc Hold’em. **In the revised manuscript, we have expanded our discussion of future directions to emphasize the importance of validating our method in richer and more realistic environments.** In future work, we plan to evaluate our approach in more complex game environments, such as multi-player or competitive-cooperative mixed settings, to further assess the generality of GOE-LLM.
> We have added this discussion in the conclusion section of the revised manuscript.
>
> ---
> ### **Response to Weakness 3**
>
> *Q: Analysis of Performance Differences Across LLM Sizes*
>
> Thank you for pointing this out.
> Figure 5 in our paper presents the performance differences across LLMs of various sizes. All LLM Exploiters were trained using the same dataset and training procedure, ensuring that performance differences can be attributed to model capacity rather than data variation. As shown in the results, the performance of the vanilla LLMs improves consistently as model size increases, which suggests that **larger models possess stronger abilities in contextual understanding and strategy learning**.
>
> Compared with the baseline, GOE-LLM already yields substantial improvements even at smaller model scales. Notably, we also observe that larger models converge faster during training, likely because **they can better fit the training data and thus learn effective strategies more efficiently.**
>
> In the revised, we have added further discussion on these observations to help readers better understand how model size affects performance, shown in 'Various LLM Sizes' of Section 4.3.

---

> ### Author Response · Authors · 2025-11-22
> **Response to Reviewer BhQn (3/3)**
>
> ---
> ### **Response to Question 2**
>
> *Q: Applicability to Multi-Agent and Non-Zero-Sum Scenarios*
>
> We thank the reviewer for raising this important point. Our current method is primarily designed for zero-sum games, and we agree that real-world multi-agent interactions often involve cooperation, coordination, and non-zero-sum payoffs. In such settings, purely focusing on opponent exploitation may indeed lead to over-optimization of adversarial behaviors and potentially cause instability.
>
> As a future direction, we plan to extend GOE-LLM to multi-agent and non-zero-sum game environments to explore how stable and effective strategy learning can be achieved under these more complex interaction dynamics. In the revised manuscript, we have added a discussion on this topic in conclusion section, highlighting the potential of applying our approach to richer multi-agent and non-zero-sum scenarios.
>
> ---
> ### **Response to Question 1 & Question 3**
>
> *Q: Stability Analysis of Mixed Best-Response Principle*
>
> We thank the reviewer for the insightful question. In our experiments, we indeed observed that arbitrarily mixing opponent strategies can introduce cyclic counter relations (as discussed in Section 3.3.1 of the main text), which may further lead to instability during training. To mitigate this issue, we adopt a mixed best-response strategy during training.
> Specifically, we **heuristically select a subset of opponent strategies for generating training data** to ensure diversity while avoiding cyclic dominance.
> As shown in Figure 2 of the main paper, Bluffing, Passive, and ValueBet form a non-transitive dominance cycle. However, by heuristically selecting only GTO, Bluffing, and Random for mixture training, we effectively avoid such cyclic relations.
>
> We further conducted 30k repeated matchups to validate the transitivity among our selected strategies:
> - GTO vs. Bluffing: as_player_0 = −0.0540, as_player_1 = 0.0636; −0.0540 + 0.0636 > 0 → **GTO > Bluffing**
>
> - GTO vs. Random: as_player_0 = 0.1663, as_player_1 = 0.1694; 0.1663 + 0.1694 > 0 → **GTO > Random**
>
> - Bluffing vs. Random: as_player_0 = 0.2204, as_player_1 = 0.1677; 0.2204 + 0.1677 > 0 → **Bluffing > Random**
>
> Thus, the final hierarchy is: **GTO > Bluffing > Random**.
>
> By choosing this transitive set of strategies, we effectively avoid non-transitive dominance cycles that could destabilize training, visulized in Figure 3 of the main paper. This design improves both the stability of the training process and the final performance of our model.
>
> ---
>
> Thanks again for your thorough and valuable review. We trust the responses above adequately address your comments. Please let us know if anything else requires clarification.

---

### Official Review · Reviewer_zdFt · 2025-10-31

**Soundness:** 3
**Presentation:** 3
**Contribution:** 2
**Rating:** 2
**Confidence:** 4

**Summary:**

This paper introduces GOE-LLM, a framework for training Large Language Model (LLM) agents to exploit opponent weaknesses in two-player zero-sum games. The approach consists of two main components: (1) an MLP-based profiler that classifies opponent strategies from behavioral history, and (2) an LLM exploiter fine-tuned using Group Relative Policy Optimization (GRPO) on a mixture of best-response strategies. The key contribution is the "Mixture-of-Best-Responses Principle," which constructs training data to avoid non-transitive dominance cycles while maintaining strategic diversity. The framework is evaluated on Kuhn Poker and Leduc Hold'em, demonstrating improved generalization to out-of-distribution opponents compared to baselines trained purely on equilibrium strategies.

**Strengths:**

1.	The problem to train an agents which can exploit any opponent is interesting.
2.	Using LLMs might be a good option due to their generalizability.

**Weaknesses:**

1.	Not well justified why LLMs are preferred? Why not trained AI models? As you still train an MLP, why not use the specific AI model for exploiter?
2.	The evaluation is not reasonable. You focus on two-player zero-sum games. However, in two-player zero-sum games, we have Nash equilibrium as the always not-lose strategy, you only need to play nash equilibrium and no need to exploit your opponent, as you will win the game. If you want to justify the reasonability, please consider more players and general-sum, where the nash equilibrium may not be a preferred strategy.

**Questions:**

1.	I want more calcification about why LLMs to be the exploiter? It seems LLM is not the necessary solution, you can use a ML model to do this. Besides, I found some related work, ICE (https://arxiv.org/abs/2408.05575) and https://arxiv.org/abs/2410.09701. They are well aligned with your scope and may not rely on LLMs, but on transformers.
2.	For the exploiting of opponents, I believe that in-context learning would be a more natural option and LLMs can do the in-context learning. and your baselines does not include in-context learning or LLMs with in-context learning. So the evaluation is not sufficient.
3.	As stated in Point 2 of weakness, two-player zero-sum games are not good testbeds for opponent exploitation where nash equilibrium is the best strategy. You need to consider general-sum or multiple player games. This also raise an issue, if you use LLMs and one of the main advantages of LLMs is their generalizability. So why not evaluating their generalizability of exploiting across games, e.g., training on Kuhn and evaluating on Leduc.
4.	The statement "does not exhibit a cyclic counter relation" is a constraint on the training data, but how is this constraint enforced during data generation? The paper doesn't provide an algorithm for constructing such datasets. The paper mentions "Final Training Size: 64,000 - 96,000" for Kuhn Poker. Why is there a range? How is the exact size determined? What is the proportion compared with the full game tree data?

---

> ### Author Response · Authors · 2025-11-22
> **Response to Reviewer zdFt (1/3)**
>
> Thanks for the detailed review and constructive feedback. Below we address the concerns you raised in your review.
>
> ---
> ### **Response to Question 1 & Weakness 1:**
>
> *Q: Why LLMs for exploitation? Related work using transformers without LLMs?*
>
> We thank the reviewer for this comment.
> Leveraging the powerful emergent capabilities of LLM, specifically in context understanding and instruction following [1], agents can rapidly adapt their strategies via context. Fundamentally, the exploiter in our GOE-LLM functions as a **language-conditioned strategic adapter**, whereas the ICE framework [2] relies on reinforcement learning from trajectories.
>
> **Our approach effectively mitigates specific limitations explicitly acknowledged in the original ICE paper[2]:**
>
> 1. While the training paradigm of ICE struggles to handle dynamic opponents, our hierarchical architecture enables the upper-level module to effectively capture and identify the opponent's strategy type based on the most recent $k$ steps. (Please refer to **Figure 6** in the Appendix, which illustrates the adaptive configuration of the $k$ parameter across tasks of varying difficulty).
> 2. Unlike ICE, which requires processing complete historical trajectories to handle complex observations, our exploiter operates more efficiently by receiving concise, high-level language descriptions of the opponent's type.
>
> In the revised manuscript, we have cited these works and explicitly highlighted these key distinctions and contributions.
>
> [1] Wei J, Tay Y, Bommasani R, et al. Emergent abilities of large language models[J]. arXiv preprint arXiv:2206.07682, 2022.
>
> [2] Li S, Yang C, Zhang Y, et al. In-context exploiter for extensive-form games[J]. arXiv preprint arXiv:2408.05575, 2024.

---

> ### Author Response · Authors · 2025-11-22
> **Response to Reviewer zdFt (2/3)**
>
> ---
> ### **Response to Question 2:**
>
> *Q: Why not evaluate LLMs with in-context learning (ICL) for exploitation?*
>
> We thank the reviewer for pointing out **that LLMs may already perform opponent exploitation through in-context learning (ICL)**. To address this concern, we conducted additional experiments evaluating GPT-4o-mini, and GPT-4o-mini augmented with our MLP Profiler.
>
> All methods were tested with the **same prompt template** as used in GOE, shown in Appendix A.3.
> To ensure statistical reliability, each model played 3,000 hands against each of 24 opponents (from 6 types), resulting in 72,000 total hands. Same as in the main experiments.
>
> The results are shown below. Bold values highlight cases that **fall below equilibrium expectations**(−1/18 as P0, +1/18 as P1):
>
> | Method          | Role | Random  | GTO(s)      | Bluff(s)    | Value(s)    | Passive(s)  | Aggressive(s) | Average  |
> |-----------------|------|---------|----------|----------|----------|----------|------------|----------|
> | GPT-4o-mini     | P0   | 0.3516  | **-0.1096**  | **-0.1112**  | -0.0504  | **-0.1165**  | **-0.1468**    | -0.0305  |
> |                 | P1   | 0.3120  | **-0.0241**  | 0.1360   | **-0.0997**  | **-0.1200**  | 0.1348     | 0.0565   |
> | GTP-4o-mini+MLP Profiler | P0   | 0.4403  | **-0.1105**  | **-0.1217**  | 0.1376   | 0.0714   | 0.0687     | 0.0809   |
> |                 | P1   | 0.4073  | **-0.0336**  | 0.1517   | 0.1221   | 0.1029   | 0.1156     | 0.1443   |
> | GOE-3B         | P0   | 0.1240  | -0.0543  | -0.0283  | 0.0733   | -0.0148  | 0.0235     | 0.0206   |
> |                 | P1   | 0.2120  | 0.0577   | 0.2185   | 0.1148   | 0.0928   | 0.1013     | 0.1328   |
>
> Key findings:
> - GPT-4o-mini does not reliably exploit opponents via ICL alone.
> - Augmenting GPT-4o-mini with our MLP Profiler significantly enhances its exploitation capabilities.
> - However, even with this enhancement, GPT-4o-mini remains less stable than GOE, particularly when acting as P0 or facing the GTO opponent. In contrast, GOE achieves stable above-equilibrium performance across all opponent types.
>
> These results support our claim that:
> 1. **Strong LLMs alone do not reliably perform opponent exploitation through ICL, and a dedicated opponent modeling mechanism is essential.**
> 2. **GOE provides a more stable and generalizable solution than simply prompting a large LLM.**
>
> Meanwhile, we also conducted experiments using GPT-5.1, playing 600 hands against each of 6 opponents (5 extreme different style opponents and 1 NE strategy), resulting in a total of 3600 hands. The results are as follows.
>
> | Method           | Role | Random | GTO(1/3) | Bluff(1,1) | Value(1) | Passive(1) | Aggressive(1) | Average |
> |------------------|------|--------|----------|------------|----------|------------|---------------|---------|
> | GPT-5.1+MLP Profiler     | P0   | 0.183  | **-0.077**   | -0.065     | 0.078    | 0.037      | **-0.162**        | -0.001  |
> |                  | P1   | 0.227  | 0.068    | 0.282      | **-0.058**   | 0.083      | 0.078         | 0.113   |
> | GOE-3B          | P0   | 0.124  | -0.059   | 0.019      | 0.086    | 0.049      | 0.091         | 0.052   |
> |                  | P1   | 0.212  | 0.062    | 0.297      | 0.154    | 0.113      | 0.135         | 0.162   |
>
> It can be observed that even with a powerful model, there are still challenges in surpassing equilibrium returns when facing extreme style opponents.

---

> ### Author Response · Authors · 2025-11-22
> **Response to Reviewer zdFt (3/3)**
>
> ---
> ### **Response to Question 3 & Weakness 2:**
>
> *Q: Why not evaluate in general-sum or multi-player games? Why not evaluate cross-game generalization?*
>
> We agree with the reviewer that general-sum or multi-player games would further strengthen the case.
> However, two-player zero-sum games remain an informative testbed for opponent exploitation, because **NE is not optimal when the goal is exploitation[1][2].**
> Playing a Nash equilibrium guarantees non-loss but does not exploit suboptimal opponents. **Our goal is precisely to investigate how much value can be extracted beyond NE play when opponent weaknesses exist[3].**
>
> And we acknowledge that cross-game generalization (e.g., training on Kuhn and evaluating on Leduc) is an interesting future direction; however, our work focuses on generalization across opponents, not generalization across tasks or game families.
>
> We have emphasized this limitation of our evaluation in N-Player, competitive-cooperative mixed game tasks in the revised conclusion section and presented it as a direction for future work.
>
> [1] Hoehn B, Southey F, Holte R C, et al. Effective short-term opponent exploitation in simplified poker[C]//AAAI. 2005, 5: 783-788.
>
> [2] Ganzfried S, Sandholm T. Game theory-based opponent modeling in large imperfect-information games[C]//The 10th International Conference on Autonomous Agents and Multiagent Systems-Volume 2. 2011: 533-540.
>
> [3] Ganzfried S, Sandholm T. Safe opponent exploitation[J]. ACM Transactions on Economics and Computation (TEAC), 2015, 3(2): 1-28.
>
> ---
> ### **Response to Question 4:**
>
> *Q: How is the "no cyclic counter relation" constraint enforced during data generation? Why the training size is a range (64k-96k)?*
>
> Thank you for highlighting this point.
> We have provided a description of the data generation process in Appendix A.2 of the revised manuscript.
>
> Regarding the dataset size, the total size of the pre-generated data differs slightly (64k samples for GOE-LLM vs. 96k samples for baseline GTO-LLM).
> Despite this discrepancy, we employed identical hyperparameters for both methods during the training phase, **specifically using a batch size of 512. Since we saved checkpoints every 100 training steps, the reported results for both models correspond to training on 51,200 samples**. This ensures that our comparison remains rigorous and is not biased by the raw size of available data.
>
> We have shown the specific data composition in Table 4 and clarified this in the revised manuscript.
>
> ---
>
> Thanks again for your thorough review and valuable comments. We hope this response provides a clear resolution to the points raised.

---

### Meta-Review · Area_Chair_9Mwe · 2026-01-05

**Summary:**

The paper proposed an LLM-based framework for learning opponent exploitation strategies. While the reviewers recognized novelty and potential appeal of the proposed approach, the overall sentiment of the reviews remained negative.

I acknowledge that the authors put a lot of effort into answering questions and addressing concerns raised by the reviewers and appreciate their active participation in the discussion and the effort to summarize the discussion. It seems to me that a lot of concerns have been addressed, while also some important and valid pieces of criticism remained as open threads:

1. **Comparison with strong non-LLM baselines.** Reviewer zdFt raised a valid point that questions whether the fact the exploiter is implemented using a finetuned LLM is essential. The authors demonstrated in the additional experiments that finetuning is important (prompting on its own doesn't work as well), but did not offer a direct comparison with strong non-LLM based methods (eg, the ICE framework, which the authors acknowledged but did not compare with). For this reason, it remains unclear how essential LLMs are for the method to work or similar results could be achieved otherwise, which is a significant weakness of the paper. It would be a much more compelling result if the the authors could demonstrate that LLMs are uniquely essential for implementing opponent exploitation strategies.

2. **Two-player zero-sum games is a suboptimal testbed for opponent exploitation.** Reviewers zdFt, BhQn, and Jxaf pointed out that broader evaluation in more complex environments (beyond Kuhn Poker and Leduc Hold’em) is necessary for clear demonstration of opponent exploitation abilities of the proposed method. I appreciate the authors' rebuttal that their evaluation is valid in the sense that they demonstrate their models are able to play better than Nash against exploitable opponents (eg, would be interested to understand how the proposed LLMs tend to behave when opponents start playing closer to Nash; are LLMs still able to find ways to exploit or things break down?). However, the selected opponents strategies are simple (as pointed by BhQn) and the overall evaluation is too poker-centric (as suggested by zdFt) for drawing broader conclusions.

Given the overall reviewer sentiment and the unaddressed important concerns, I recommend rejecting the paper. Again, I very much appreciate the authors actively participating in the discussion and hope it would be useful for them to further improve the paper before resubmitting elsewhere.

**Reviewer Concerns:**

**Addressed concerns:**
- Added comparison with in-context learning (or prompting-based) baselines that use large scale LLMs (requested by zdFt, BhQn, RXu2)
- Added missing citations and related work.
- Addressed all clarifications questions.

**Outstanding concerns:** as mentioned in the summary above, (1) comparison with strong non-LLM baselines to confirm importance of using LLMs for opponent exploitation and (2) expanding the scope beyond two-player zero-sum games to better support the claim about generalized opponent exploitation remained unaddressed.

**Reviewer Scores:**

zdFt: original score was 2, which the reviewer might have increased to 3-4 based on discussion.
Jxaf: indicated that they would maintain their original score of 6.
BhQn: original score was 6, which they would likely maintain (similar to Jxaf) given the discussion.
RXu2: original score was 2, which the reviewer might have increased to 3-4 based on the fact that most of their questions were addressed and/or were stemming from some misunderstanding.

---

### Decision · Program_Chairs · 2026-01-26

Reject